# TRAINING-FREE REWARD-GUIDED IMAGE EDITING VIA TRAJECTORY OPTIMAL CONTROL

**Jinho Chang**[*]**, Jaemin Kim**[*]**& Jong Chul Ye**
Graduate School of Artificial Intelligence
Korea Advanced Institute of Science and Technology
Seoul, South Korea
{jinhojsk515,kjm981995,jong.ye}@kaist.ac.kr

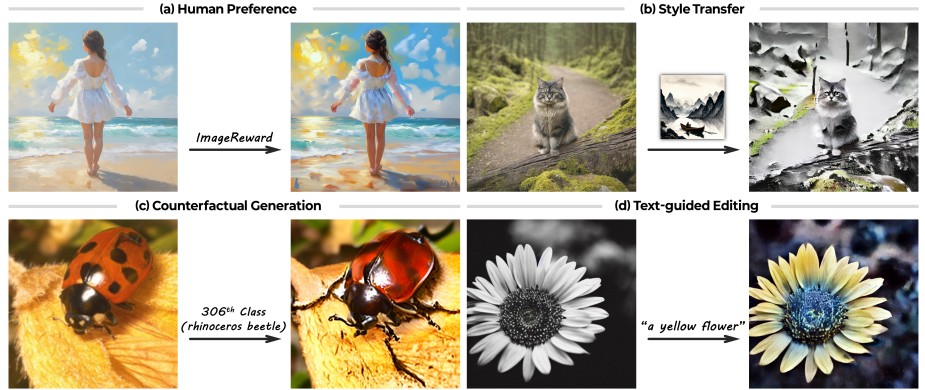

Figure 1: **Reward-guided image editing samples with unconditional diffusion and flow-matching models.** Reward-guided edited samples across various tasks, such as (a) Human preference, (b) Style transfer, (c) Counterfactual generation, and (d) Text-guided image editing.

## ABSTRACT

Recent advancements in diffusion and flow-matching models have demonstrated remarkable capabilities in high-fidelity image synthesis. A prominent line of research involves reward-guided guidance, which steers the generation process during inference to align with specific objectives. However, leveraging this reward-guided approach to the task of image editing, which requires preserving the semantic content of the source image while enhancing a target reward, is largely unexplored. In this work, we introduce a novel framework for training-free, reward-guided image editing. We formulate the editing process as a trajectory optimal control problem where the reverse process of a diffusion model is treated as a controllable trajectory originating from the source image, and the adjoint states are iteratively updated to steer the editing process. Through extensive experiments across distinct editing tasks, we demonstrate that our approach significantly outperforms existing inversion-based training-free guidance baselines, achieving a superior balance between reward maximization and fidelity to the source image without reward hacking.

## 1 INTRODUCTION

Following the advancement of diffusion and flow-matching models that led to remarkable success in high-fidelity image synthesis (Ho et al., 2020; Dhariwal & Nichol, 2021; Lipman et al., 2022), various methods have been developed to edit real-world images (Meng et al., 2021; Hertz et al., 2023) by leveraging their pre-trained image priors. However, most editing techniques remain limited to concepts that exist within the model's pre-trained distribution (*i.e.*, applying a *"Van Gogh style"* is only possible if the model has been trained in such styles). While text-to-image models (Rombach et al., 2022; Esser et al., 2024) provide diverse conditional distributions, abstract human preferences or subtle stylistic nuances are often difficult to specify clearly using natural language.

---

[*]Equal contribution to this work.

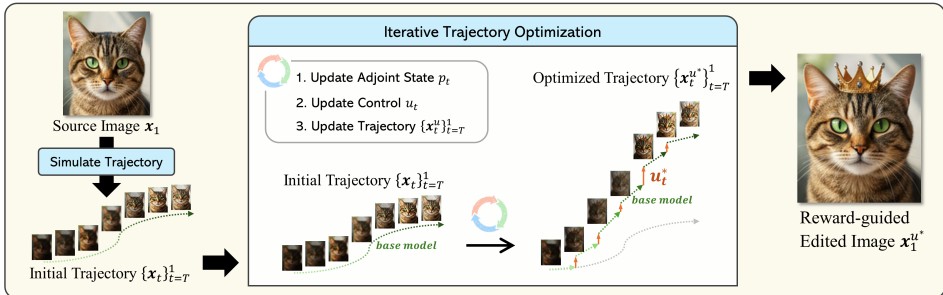

Figure 2: **Methodology overview.** Given a source image $x_1$, our method first generates its corresponding initial trajectory. We then progressively refine this trajectory by solving a reward-guided optimal control problem. This process steers the path into an optimized trajectory, whose endpoint is the final edited image $x_1^{u^*}$.

Meanwhile, reward-guided sampling methods have been proposed as a promising, training-free framework that operates during inference (Chung et al., 2023; Yu et al., 2023; Song et al., 2023; Ye et al., 2024; Geng & Owens, 2024), which leverages off-the-shelf differentiable reward functions to steer the generation process toward a desired objective. The primary advantage of this approach is its ability to generate images toward a novel target distribution defined by the reward function, moving beyond the original sample distribution.

Despite the progress of reward-guided image generation, its potential for image editing techniques has been under-explored, and there is room for improvement. Reward-guided editing is more challenging, as it requires both maximizing a reward and preserving the core identity of the source image. The most intuitive approach is to first invert the source image into the noise space and then apply a reward-guided generation algorithm during the reverse process. Unfortunately, this method often fails because most guidance techniques rely on the reward gradient to the intermediate noised image or one-step approximation of the clean image, but for complex and non-linear reward functions, this indirect guidance degrades the structural faithfulness of the source image (Chung et al., 2023; Yu et al., 2023; Ye et al., 2024).

To address these challenges, here we propose a training-free framework by reformulating reward-guided image editing as a trajectory optimal control problem (Figure 2). Specifically, we treat the reverse diffusion process, originating from the source image, as a controllable trajectory. Our goal is then to find the optimal control signal that steers this entire trajectory to a terminal state that maximizes the reward. To solve this control problem, we develop an iterative adjoint-state update algorithm based on the principles of Pontryagin's Maximum Principle (PMP) (Levine, 1972). We comprehensively demonstrate the effectiveness of our approach across four distinct editing tasks (Figure 1). By optimizing the entire path, our approach shows that the resulting edits are not only effective in terms of the target reward but also structurally coherent with the source image. Our main contributions are threefold:

- We present a training-free reward-guided image editing framework by formulating it as a trajectory optimal control problem, applicable to both diffusion and flow-matching models.
- Based on the PMP necessary conditions, we develop an iterative adjoint-state optimization procedure to find the optimal trajectory that maximizes the target reward.
- Through extensive experiments across diverse tasks, we demonstrate that our method outperforms existing inversion-based guidance baselines, achieving superior results without reward hacking or structural degradation.

## 2 RELATED WORKS

### 2.1 TRAINING-FREE IMAGE EDITING WITH DIFFUSION AND FLOW-MATCHING MODELS

The exploitation of the pre-trained distribution from pre-trained models enabled various techniques for the image editing task. One of the most popular approaches is inversion-based methods (Meng et al., 2021; Mokady et al., 2023; Huberman-Spiegelglas et al., 2024), which map a source image

to noise space through the forward process and then edit it through a modified reverse trajectory. Another direction employs distillation-based optimization (Poole et al., 2023; Hertz et al., 2023; Nam et al., 2024), which guides the source image without an explicit sampling step. Some used empirical feature alignment, such as cross-attention map (Hertz et al., 2022; Cao et al., 2023), to ensure the sampling output retains the source image feature. More recently, flow-matching approaches such as FlowEdit (Kulikov et al., 2024) achieve optimization-free editing by directly steering text-conditional flows. Leveraging the semantic coverage of large-scale text-to-image models like Stable Diffusion (Rombach et al., 2022; Esser et al., 2024), editing is typically specified with natural language prompts. However, these approaches are restricted by the scope of the model's pre-trained distribution, making it difficult to edit beyond the concepts it has trained.

## 2.2 REWARD-GUIDED IMAGE GENERATION

Recently, modifying the generative process to align with a user-defined objective, often encapsulated by a reward function, is a central goal in controllable generation. While this can be done by explicit training for the reward-aligned distribution (Black et al., 2023; Wallace et al., 2024), several works aim to apply training-free guidance that steers the sampling process (Chung et al., 2023; Ye et al., 2024; Yu et al., 2023; He et al., 2023; Song et al., 2023). Leveraging off-the-shelf differentiable predictors (*e.g.*, a classifier or a reward model), these approaches modify the denoising samples or their posterior mean during inference to achieve a higher reward at the end of the sampling. Nevertheless, their potential for editing has been underexplored since these methods were fundamentally designed for sampling from the noise distribution.

## 2.3 STEERING GENERATIVE MODELS WITH OPTIMAL CONTROL

Leveraging the iterative sampling process of diffusion and flow-matching models, recent works (Rout et al., 2024; 2025; Zhu et al., 2025) have employed optimal control perspectives to modify the sampling trajectory to satisfy certain desired properties such as style personalization, generalized Doob's *h*-transform, and inversion proximal to the given endpoint. For reward-alignment model training, Adjoint Matching (Domingo-Enrich et al., 2025) formulated a Stochastic Optimal Control (SOC) problem, where the goal is to maximize the terminal reward while regularizing the control term. While optimal control has been successfully applied to model fine-tuning and sampling, its application to the training-free editing task has been relatively under-explored. Our work is to adapt the principles of the trajectory optimal control problem for editing a given source image, steering its sampling trajectory towards a target reward without model updates.

# 3 PRELIMINARIES

## 3.1 DIFFUSION AND FLOW-MATCHING MODELS

**Diffusion Models.** Diffusion models (Dhariwal & Nichol, 2021; Song et al., 2021b) are a class of generative models trained to reverse the predefined forward process that gradually injects Gaussian noise into clean data $\boldsymbol{x}_1$ over a time interval $t \in [0,1]$[1]. The diffusion model $\boldsymbol{\epsilon}_\theta$ is trained by a denoising score matching (DSM) objective (Vincent, 2011; Ho et al., 2020) to predict the injected noise from the perturbed sample $\boldsymbol{x}_t \sim \mathcal{N}(\sqrt{\bar{\alpha}_t}\boldsymbol{x}_1, 1 - \bar{\alpha}_t\boldsymbol{I})$ where $\{\bar{\alpha}_t\}_{t=0}^1$ is a set of parameters to control the noise level. The reverse sampling to generate $\boldsymbol{x}_{t+dt}$ from $\boldsymbol{x}_t$ using the following,

$$\boldsymbol{x}_{t+dt} = \frac{\sqrt{\bar{\alpha}_{t+dt}}\left(\boldsymbol{x}_t - \sqrt{1-\bar{\alpha}_t}\boldsymbol{\epsilon}_\theta(\boldsymbol{x}_t,t)\right)}{\sqrt{\bar{\alpha}_t}} + \sqrt{1 - \bar{\alpha}_{t+dt} - \sigma_t^2}\boldsymbol{\epsilon}_\theta(\boldsymbol{x}_t,t) + \sigma_t\boldsymbol{\epsilon}, \quad \boldsymbol{\epsilon} \sim \mathcal{N}(0,\boldsymbol{I}), \quad (1)$$

where $\sigma_t$ controls the stochasticity (Song et al., 2021a).

**Flow-matching Models.** Flow-matching models (Lipman et al., 2022; Liu et al., 2022; Esser et al., 2024) define their sampling processes through interpolating between a known prior and the target data distribution. The sampling process is typically governed by an Ordinary Differential Equation (ODE) over the time interval $[0,1]$:

$$d\boldsymbol{x}_t = \boldsymbol{v}_\theta(\boldsymbol{x}_t,t)dt, \quad \boldsymbol{x}_0 \sim \mathcal{N}(0,\boldsymbol{I}). \quad (2)$$

---

[1]Instead of the notation typically used in diffusion models, we employ the notation used in flow-matching models, where the timestep $t$ spans from 0 (noise) to 1 (data) with evenly spaced interval.

The parameterized velocity field $\boldsymbol{v}_\theta(\boldsymbol{x}_t, t)$ is trained to approximate the marginal derivative of a pre-defined reference flow across the training data, typically of the form $\boldsymbol{x}_t = \beta_t \boldsymbol{x}_0 + \alpha_t \boldsymbol{x}_1$ with $(\alpha_t, \beta_t)$ satisfying boundary conditions $\alpha_0 = \beta_1 = 0$ and $\alpha_1 = \beta_0 = 1$. The most common setting lets $\alpha_t = t$ and $\beta_t = 1 - t$. This training objective ensures that the solution of the sampling ODE has the same marginal distributions as the reference flow, thereby guaranteeing that $\boldsymbol{x}_1$ follows the target data distribution.

**Unified SDE Framework.** Although diffusion and flow-matching model originates from different theoretical foundations, their sampling processes can be unified with a Stochastic Differential Equation (SDE). Leveraging the SDE perspective of the diffusion reverse process (Song et al., 2021b) and the Fokker-Planck equation, the sampling dynamics for both models can be expressed as:

$$d\boldsymbol{x}_t = b(\boldsymbol{x}_t, t)dt + \sigma_t d\mathbf{B}_t, \quad \boldsymbol{x}_0 \sim \mathcal{N}(0, \mathbf{I}), \tag{3}$$

where $b(\boldsymbol{x}_t, t)$ is the drift term, $\sigma_t$ is an arbitrary time-dependent diffusion coefficient, and $d\mathbf{B}_t$ is a Brownian motion. With the diffusion model scheduler $\{\bar{\alpha}_t\}_{t=0}^1$ and flow-matching model setting of $\alpha_t = t$ and $\beta_t = 1 - t$, the drift term can be further specified as (Domingo-Enrich et al., 2025):

$$b_{\text{Diffusion}}(\boldsymbol{x}_t, t) = \frac{\dot{\bar{\alpha}}_t}{2\bar{\alpha}_t}\boldsymbol{x}_t - \left(\frac{\dot{\bar{\alpha}}_t}{2\bar{\alpha}_t} + \frac{\sigma_t^2}{2}\right)\frac{\boldsymbol{\epsilon}_\theta(\boldsymbol{x}_t, t)}{\sqrt{1 - \bar{\alpha}_t}} \tag{4}$$

$$b_{\text{Flow-Matching}}(\boldsymbol{x}_t, t) = \boldsymbol{v}_\theta(\boldsymbol{x}_t, t) + \frac{t\sigma_t^2}{2(1-t)}\left(\boldsymbol{v}_\theta(\boldsymbol{x}_t, t) - \frac{1}{t}\boldsymbol{x}_t\right), \tag{5}$$

where $\dot{\bar{\alpha}}_t$ denotes $\frac{d\bar{\alpha}}{dt}$. Under this framework, diffusion models correspond to particular choices of $(\bar{\alpha}_t, \sigma_t)$ that recover the DDIM samplers (Song et al., 2021a), while flow-matching models are recovered in the deterministic limit $\sigma_t = 0$ or its stochastic extension. This unified formulation allows us to analyze and manipulate both model types using a single theoretical perspective, and provides a way for control-theoretic interventions.

## 3.2 OPTIMAL CONTROL PROBLEM

Optimal control (OC) is a mathematical framework for finding an optimal strategy to steer a dynamical system to minimize cost functional. While OC encompasses a wide range of problem formulations, we focus on the quadratic cost and additive control problem, starting from the given initial state $\boldsymbol{x}_0 \in \mathbb{R}^d$. Consider continuous-time dynamics at Eq. (3), where $b : \mathbb{R}^d \times [0, 1] \rightarrow \mathbb{R}^d$ and $\sigma_t \in \mathbb{R}$. The OC problem aims to find the additional optimal control term $u : \mathbb{R}^d \times [0, 1] \rightarrow \mathbb{R}^d$ that minimizes the following cost functional [2]:

$$\min_{u \in \mathcal{U}} \mathbb{E}\left[\int_0^1 \left(\tfrac{1}{2}\|u(\boldsymbol{x}_t^u, t)\|^2 + f(\boldsymbol{x}_t^u, t)\right) dt + g(\boldsymbol{x}_1^u)\right] \tag{6}$$

$$\text{s.t.} \quad d\boldsymbol{x}_t^u = \left(b(\boldsymbol{x}_t^u, t) + \sigma_t u(\boldsymbol{x}_t^u, t)\right) dt + \sigma_t d\mathbf{B}_t, \quad \boldsymbol{x}_0^u = \boldsymbol{x}_0$$

where $f$ is the running cost and $g$ is the terminal cost. This optimal control problem has been extensively studied in both deterministic and stochastic settings, and analytical tools such as the Hamilton–Jacobi–Bellman (HJB) equations (Fleming & Rishel, 2012) and Pontryagin's Maximum Principle (PMP) (Levine, 1972) provide necessary and, in some cases, sufficient conditions for optimality.

# 4 METHODS

## 4.1 MOTIVATION: FROM GRADIENT ASCENT TO TRAJECTORY CONTROL

Assuming a differentiable reward function $r(\cdot)$, the most direct approach for editing a given image $\boldsymbol{x}_1$ to maximize $r(\cdot)$ is to perform Gradient Ascent (GA) in the pixel space. While this provides the steepest direction to optimize the image, it disregards the underlying image prior, leading to adversarial and out-of-distribution results that are perceptually unrealistic (Goodfellow et al., 2014). An alternative to prevent this is to leverage the generative model's prior, by first performing deterministic inversion into noise space (Mokady et al., 2023) and then applying reward-guided sampling

---

[2]The expectation over the Brownian motion in Eq. (6) can be removed by setting $\sigma_t = 0$, or fixing the Brownian motion with a certain realization of $\sigma_t d\mathbf{B}_t$ as a constant.

methods during the reverse process. However, reward-guided sampling is fundamentally constrained by its reliance on approximated guidance; since any noiseless image is not available in the sampling process, samples are optimized to increase the reward on the posterior mean (Efron, 2011) of clean images from the given noised image. As the reward function becomes more complex and non-linear, this guidance can be ineffective or even corrupt the global consistency of the image structure. Moreover, previous guided sampling methods cannot provide a theoretical justification for the selection of the guidance scale, and require careful hyperparameter tuning to find their optimal performance.

To overcome these limitations, we propose a novel image editing methodology for the guidance term that is both effective and minimizes off-manifold phenomenon, by rephrasing the problem as the optimization of the entire generation trajectory with optimal control.

## 4.2 PROBLEM FORMULATION

Let's say $\{x_t\}_{t=T}^1$ is given as an initial trajectory sampled from Eq. (3), where $x_1$ denotes the given source image and $T \in [0, 1)$ is the starting noise depth. Even for real-world images that were not generated by the model, there are methods to get an initial trajectory $\{x_t\}_{t=T}^1$ that ends at the given image, which are further discussed in Section 4.3. Our goal is to introduce an additional control term $u_t^*$ into the drift and find the optimized trajectory $\{x_t^{u^*}\}_{t=T}^1$ that still starts from $x_T$ but produces an edited image $x_1^{u^*}$ that remains realistic and faithful to a source image $x_1$ while maximizing the reward $r(\cdot)$. Formally, we solve the following optimal control problem,

$$\min_{u \in \mathcal{U}} \int_T^1 \tfrac{1}{2}\|u(x_t^u, t)\|^2 dt - r(x_1^u) \tag{7}$$
$$\text{s.t.} \quad dx_t^u = (b(x_t^u, t) + u(x_t^u, t))dt + \sigma_t d\mathbf{B}_t, \quad x_T^u = x_T,$$

where the Brownian component will be replaced by the fixed realization according to the given $\{x_t\}_{t=T}^1$ since we only focus on the optimization of the single trajectory. Since both the base drift term and reward functions are complex and non-linear, it is impractical to find a closed-form solution for $u(x_t^u, t)$ that guarantees the global minimum of the cost. Nonetheless, PMP states the necessary condition that the optimal control term of Eq. (7) satisfies. Specifically, by introducing a Hamiltonian $\mathcal{H}(x_t, u, p_t, t) = p_t^\top (b(x_t, t) + u) + \frac{1}{2}\|u\|^2$, where $p_t$ is often called the adjoint state, the optimal trajectory satisfies three coupled differential equations:

$$\frac{dx_t^*}{dt} = \nabla_{p_t}\mathcal{H}(x_t^*, u^*, p_t, t) = b(x_t^*, t) + u_t^*, \qquad x_T^* = x_T \tag{8}$$

$$\frac{dp_t^*}{dt} = \nabla_{x_t}\mathcal{H}(x_t, u^*, p_t^*, t) = -\left[\nabla_{x_t}b(x_t^*, t)\right]^\top p_t^*, \qquad p_1^* = -\nabla_{x_1}r(x_1^*) \tag{9}$$

$$u_t^* = \arg\min_{u \in \mathcal{U}} \mathcal{H}(x_t^*, u, p_t^*, t) = -p_t^* \tag{10}$$

Therefore, our goal is to find the optimal control $u^*$ to construct the trajectory that satisfies these optimality conditions. After we find the optimized trajectory $\{x_t^{u^*}\}_{t=T}^1$, we take the terminal point $x_1^{u^*}$ as a reward-guided editing result of $x_1$. Compared to Adjoint Matching (Domingo-Enrich et al., 2025), which had to formulate its goal into a *stochastic* optimal control problem to fine-tune the entire model's marginal distribution, our formulation directly targets the single-image editing.

## 4.3 ITERATIVE TRAJECTORY OPTIMIZATION VIA ADJOINT GUIDANCE

However, jointly optimizing $x_t$, $u_t$, and $p_t$ across all time steps is computationally impractical. Therefore, we propose an iterative approach analogous to Coordinate Descent (Wright, 2015). In each iteration, we sequentially update each component to better satisfy the PMP conditions:

1. **Compute Adjoint State** $p_t$: With the current trajectory and control $\{x_t, u_t\}_{t=T}^1$ fixed, we solve the adjoint equation Eq. (9) backward in time to compute the adjoint states $\{p_t\}_{t=T}^1$.

2. **Update Control** $u_t$: We then update the control $\{u_t\}_{t=T}^1$ towards $-p_t$, according to the optimality condition of Eq. (10).

3. **Update Trajectory** $x_t$: With the updated control, we simulate a new, updated trajectory $\{x_t\}_{t=T}^1$ using Eq. (8).

---

**Algorithm 1** Image Editing via Trajectory Optimization Control

---

**Require:** Source image $\boldsymbol{x}_1$, Depth $0 < T < 1$, Number of iteration $N$, Unconditional base model $\theta$, Learning rate $\lambda$, Reward function $r(\cdot)$, Reward weight $w$

1: $\{\boldsymbol{x}_t\}_{t=T}^1, \{\boldsymbol{B}_t\}_{t=T}^1 = \texttt{simulate\_trajectory}(\boldsymbol{x}_1, \theta)$
2: $\{u_t\}_{t=T}^1 := \boldsymbol{0}$
3: **for** $iter = 1$ **to** $N$ **do**
4: $\quad \{p_t\}_{t=T}^1 = \texttt{compute\_adjoint}(\{\boldsymbol{x}_t\}_{t=T}^1, \theta, wr(\cdot))$  $\quad\quad\quad\quad$ ▷ Compute $p_t$ from current $\boldsymbol{x}_t$
5: $\quad u_t = u_t - \lambda(u_t + p_t)$ **for** $t = 1, ..., T$  $\quad\quad\quad\quad\quad\quad\quad\quad$ ▷ Update $u_t$ towards $-p_t$
6: $\quad \boldsymbol{x}_{t+dt} = \boldsymbol{x}_t + \{b_\theta(\boldsymbol{x}_t, t) + u_t\}dt + \boldsymbol{B}_t$ **for** $t = T, ..., 1 - dt$  $\quad$ ▷ Get $\boldsymbol{x}_t$ with updated $u_t$
7: **end for**
8: **return** $\boldsymbol{x}_1$

---

This iterative process is repeated, progressively refining the trajectory until it converges to a path that locally satisfies the optimality conditions, yielding a final edited image $\boldsymbol{x}_1^{u^*}$ that achieves a higher reward while maintaining high fidelity to the source image, as illustrated in Figure 2.

Algorithm 1 describes the proposed image trajectory optimization process. We denote the function that generates the initial image trajectory as $\texttt{simulate\_trajectory}$. For our primary results, we utilized deterministic DDIM inversion for diffusion models and the time-reversed ODE for flow-matching models, as a noiseless trajectory with $\sigma_t = 0$. We discuss alternative stochastic methods for initial trajectory generation in Section 6. Note that compared to previous methods with empirical guidance scale search (Ye et al., 2024), the guidance scale in all steps can be controlled by a weight parameter $w$ on the terminal reward function $r(\cdot)$. More specified algorithms for diffusion and flow-matching models are detailed in Appendix A.1. Furthermore, we discuss the advantage of our method over the previously suggested guided sampling methods in Appendix B.2, by the link between their guidance terms and the optimal control problem.

## 5 EXPERIMENTS

In this section, we evaluated our method and several baselines to edit the given images to improve the desired reward. We designed four scenarios with different reward objectives: Human Preference, Style Transfer, Counterfactual Generation, and Text-guided Image Editing.

### 5.1 EXPERIMENTAL SETUP

**Models and Baselines.** We used StableDiffusion 1.5 (Rombach et al., 2022) and StableDiffusion 3 (Esser et al., 2024) as our primary unconditional diffusion and flow-matching model, respectively. The main results are reported using the diffusion model, while the results for the flow-matching model are provided in Appendix B.1. We compared our method against two categories of baselines: Naive Gradient Ascent (GA), which directly adds the reward gradient to the source image. Second, we adapt an image inversion followed by several reward-guided sampling methods, including DPS (Chung et al., 2023), FreeDoM (Yu et al., 2023), and TFG (Ye et al., 2024). For all experiments, we only utilized unconditional models to isolate the effect of the reward guidance from any text conditioning. Detailed hyperparameter settings for each method are provided in Appendix A.2.

**Datasets.** We prepared a diverse set of datasets depending on the task: Images sampled with the prompts from REFL (Xu et al., 2024) for human preference, Pick-a-Pic (Kirstain et al., 2023) for style transfer, ImageNet-1k (Deng et al., 2009) for counterfactual generation, and CelebA-HQ (Karras et al., 2017) for text-guided facial editing. Each evaluation is performed on 300 randomly selected images from the respective datasets.

**Evaluation Metrics.** Our metrics are designed to quantify three aspects: (1) Effectiveness of the method to increase the target reward. (2) The output's generalizability beyond target reward overfitting, which we measured with different reward functions for the same quality. (3) Preservation of the content and structure of the source image, which we mostly employed LPIPS (Zhang et al., 2018) and CLIP cosine similarity (Radford et al., 2021) between the source and edited images (CLIP-I$_{src}$).

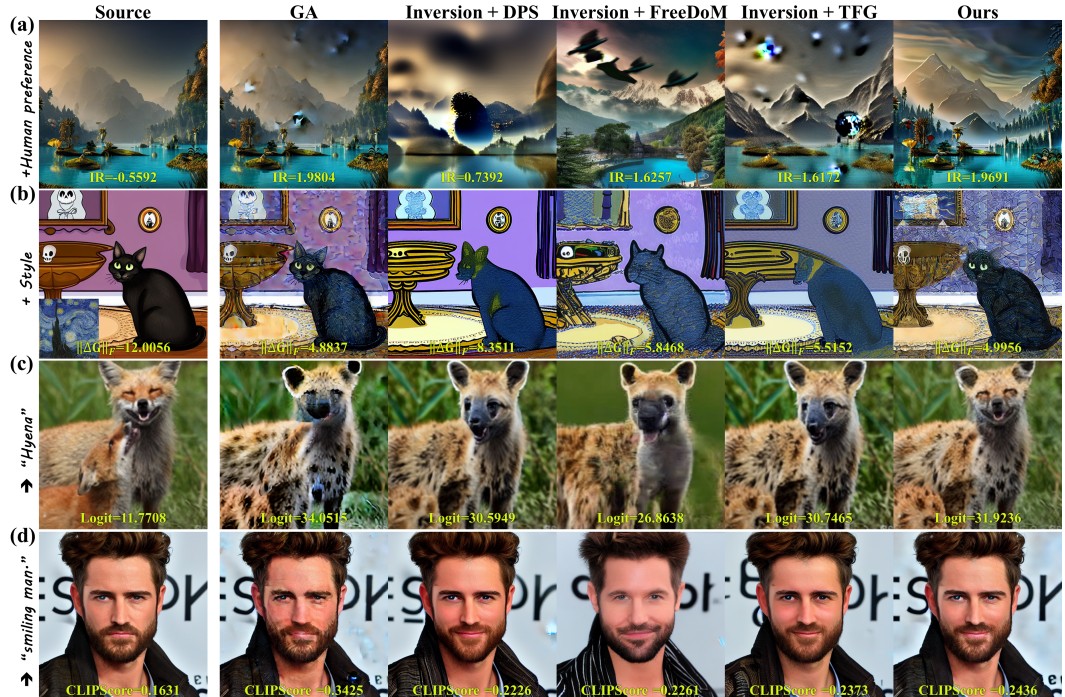

Figure 3: **Qualitative comparison** on (a) Human preference, (b) Style transfer, (c) Counterfactual generation, and (d) Text-guided image editing. Each image's target reward is written in yellow.

| Method | Target reward ImageReward[↑] | Validation metrics | | | Source preservation | |
|---|---|---|---|---|---|---|
| | | HPSv2[↑] | CLIPScore[↑] | Aesthetic[↑] | LPIPS[↓] | CLIP-I$_{src}$[↑] |
| None | 0.1542 | 0.2385 | 0.2887 | 6.0516 | 0.0000 | 1.0000 |
| Gradient Ascent | **1.9088** | 0.2247 | 0.2877 | 5.5775 | **0.1474** | 0.9195 |
| Inversion+DPS | 1.5988 | 0.2323 | 0.2650 | 5.8276 | 0.2875 | 0.8505 |
| Inversion+FreeDoM | 1.5995 | 0.2226 | 0.2356 | 5.4951 | 0.5503 | 0.7225 |
| Inversion+TFG | 1.7053 | 0.2362 | 0.2727 | 5.6331 | 0.2927 | 0.8401 |
| Ours | 1.8914 | **0.2526** | **0.2904** | **6.1088** | 0.1717 | **0.9242** |

Table 1: Quantitative results for higher human preference. **Bold**: best, underline: second best.

## 5.2 RESULTS

We discuss the performance of our method across different scenarios, where several examples and the qualitative comparison with baselines are shown in Figure 1 and Figure 3, respectively.

**Human Preference.** Human preference captures a composite concept of image quality, prompt alignment, and other subjective factors. Although it's difficult to express through explicit conditions such as text, several proxy metrics have been proposed. We adopt the ImageReward (Xu et al., 2024) between the image and its corresponding text prompt as the target reward function, which is trained to predict human preference scores. We evaluate HPSv2 (Wu et al., 2023), image-text CLIP-Score (Radford et al., 2021), and Aesthetic Score (Schuhmann, 2022) as similar validation metrics for the generalizability.

The first row of Figure 3 shows a qualitative comparison across methods. GA leaves the source image mostly unchanged while introducing severe artifacts, which are a clear indication of reward hacking. This is further shown in Table 1, where GA achieves the highest target reward, but its generalization to other human preference metrics is limited. Meanwhile, guided-sampling-based methods deviate the image more than ours through their sampling process, which doesn't regard the source image. Moreover, the result often has severe structural degradation. This stems from the importance of high-frequency details of the reward function, where the guidance on the blurred posterior mean can be ineffective or harmful. In contrast, our approach achieves better target reward and source image fidelity than guided sampling baselines, with a generalized performance that also

| Method | Target reward $\|\Delta G\|_F[\downarrow]$ | Validation metrics CLIP-I$_{sty}$ [↑] | DINO$_{sty}$[↑] | Source preservation CLIP-I$_{src}$[↑] |
|---|---|---|---|---|
| None | 12.190 | 0.4757 | 0.1236 | 1.0000 |
| Gradient Ascent | **4.8742** | 0.5270 | 0.1953 | **0.8374** |
| Inversion+DPS | 6.8435 | 0.5395 | 0.1693 | 0.6858 |
| Inversion+FreeDoM | 5.4619 | 0.5629 | 0.2250 | 0.6207 |
| Inversion+TFG | 6.2641 | 0.5455 | 0.1938 | 0.7076 |
| Ours | 5.0185 | **0.5782** | **0.2467** | 0.7169 |

Table 2: Quantitative results on style transfer. $\Delta G$ denotes the difference between the Gram matrix of the editing output and the style reference image. **Bold**: best, underline: second best.

| Method | Target reward Logit$_{tgt}$[↑] | Validation metrics CLIPScore [↑] | Source preservation LPIPS[↓] | CLIP-I$_{src}$[↑] |
|---|---|---|---|---|
| None | 4.8722 | 0.1452 | 0.0000 | 1.0000 |
| Gradient Ascent | **24.875** | 0.1908 | 0.2246 | **0.8483** |
| Inversion+DPS | 20.378 | 0.1811 | 0.3251 | 0.7305 |
| Inversion+FreeDoM | 17.891 | 0.1736 | 0.4801 | 0.6411 |
| Inversion+TFG | 18.854 | 0.1757 | 0.2972 | 0.7607 |
| Ours | 23.372 | **0.1936** | **0.2251** | 0.8256 |

Table 3: Quantitative results on counterfactual generation. **Bold**: best, underline: second best.

increases the validation metrics. This suggests that by optimizing the entire trajectory, our method avoids reward hacking and produces more coherent, high-quality editing results.

**Style Transfer.** The goal is to edit a source image with the artistic style of a reference image while preserving its original content. The target reward is defined as the negated Frobenius norm of the Gram matrix difference ($\|\Delta G\|_F$) extracted from the edited image and the reference. Style reference images were selected from Hertz et al. (2024). Following previous works on style transfer (Rout et al., 2024; Hertz et al., 2024), style alignment is validated with CLIP cosine similarity (CLIP-I$_{sty}$) and DINO cosine similarity (DINO$_{sty}$) (Caron et al., 2021) between the output and the style reference image. The source image preservation is only measured by CLIP-I$_{src}$ since LPIPS wasn't instructive for the task that changes the entirety of the image.

Again, our method achieves the highest validation metrics as shown in Table 2, while the effect of GA is only limited to its target reward. Guided sampling-based methods unavoidably distort the source image's content in the process of stylization. The second row of Figure 3 illustrates that our trajectory optimization offers both stylistically faithful and structurally coherent images.

**Counterfactual Generation.** Counterfactuals are widely used in explainable AI, as they reveal what minimal changes are sufficient to alter the decision of the classifier, offering human-interpretable insights into the model's reasoning (Verma et al., 2024; Kim et al., 2025b). In this section, we edit the image to alter the classifier's decision with minimal structural change. Using a pre-trained robust classifier on ImageNet-1k (Santurkar et al., 2019), we define the reward as the logit value (Logit$_{tgt}$) of a new target class different from the source image. The target class was selected to be close to the original class based on the Bostock (2019) ImageNet-1k hierarchy. We use the CLIPscore for the generalizability, with the text prompt of *"a photo with* [class]*"*.

As presented in Table 3 and the third row of Figure 3, our method effectively generates counterfactual examples by sufficiently increasing the target class logit while preserving the overall appearance of the image. Note that GA shows better validation metrics and image quality compared to other baselines in this task, only because the reward function is highly robust to adversarial attacks. In contrast, our method achieves better or comparable reward optimization and source image preservation throughout various tasks, without any assumptions or restrictions on the objectives.

**Text-guided Image Editing.** Unlike most approaches that rely on models trained to learn a text-conditional distribution, we frame the classic task of text-guided editing within our reward-based framework. Following prior work on reward-driven text-based editing (Liu et al., 2023), we design our scenario on the CelebA-HQ (Karras et al., 2017) dataset. We use CLIPScore between the edited image and a target text prompt (e.g., *"A smiling man."*) as a reward. The target text prompts are randomly generated for each image to change one of its features according to the CelebA-HQ attributes (Na et al., 2022). We additionally use ImageReward and HPSv2 as separate metrics to evaluate image-text alignment.

| Method | Target reward CLIP[↑] | Validation metrics | | Source preservation | |
| --- | --- | --- | --- | --- | --- |
| | | ImageReward[↑] | HPSv2 [↑] | LPIPS[↓] | CLIP-$I_{src}$[↑] |
| None | 0.1760 | -0.2404 | 0.2233 | 0.0000 | 1.0000 |
| Gradient Ascent | **0.3567** | -0.2331 | 0.2193 | **0.1250** | **0.6660** |
| Inversion+DPS | 0.3173 | -0.2923 | 0.2032 | 0.3658 | 0.5300 |
| Inversion+FreeDoM | 0.3158 | -0.1100 | 0.2094 | 0.4492 | 0.5147 |
| Inversion+TFG | 0.3260 | -0.2801 | 0.2040 | 0.3745 | 0.5282 |
| Ours | 0.3441 | **0.0976** | **0.2243** | 0.2252 | 0.6280 |

Table 4: Quantitative results on text-guided image editing. **Bold**: best, underline: second best.

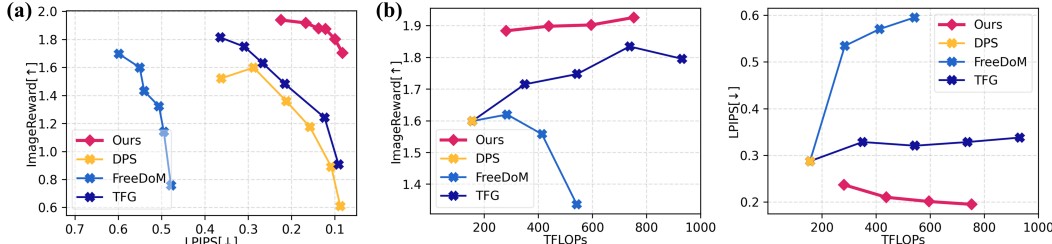

Figure 4: (a) Trade-off between target reward and source image fidelity with different guidance scale hyperparameters. (b) Evolvement of the target reward and source image fidelity with increasing computational cost.

Our approach achieves the best alignment with the textual description in both quantitative measures (Table 4) and perceptually (the last row of Figure 3). While inversion-based sampling methods can also produce appropriate results, they inevitably lose more of the information from the source images (*e.g.*, the letters in the background), leading to lower LPIPS and CLIP-$I_{src}$.

**User Study.** To validate the perceptual quality, we conducted a user study with 42 different participants who were asked across different categories: alignment with the target reward, faithfulness to the source image, and quality of the edited image. Each participant viewed 50 images and rated them on a 5-point scale. The results on the Table 5 demonstrated that our model significantly outperformed the baseline models in terms of perceptual quality.

| Method | Reward align. | Faith-fulness | Quality |
| --- | --- | --- | --- |
| Gradient Ascent | 2.75 | 3.37 | 2.53 |
| Inv. + DPS | 3.28 | 3.15 | 3.12 |
| Inv. + FreeDoM | 2.90 | 2.45 | 2.42 |
| Inv. + TFG | 3.02 | 2.94 | 2.74 |
| Ours | **3.67** | **3.60** | **3.36** |

Table 5: User study result.

## 6 DISCUSSION

**Reward-Fidelity Tradeoff with Different Guidance Scale.** While our method has been shown to provide superior editing performance across various scenarios, the inherent trade-off between reward alignment and source fidelity is present in all guidance-based approaches. For a fairer comparison, we evaluated the performance of our method and the baselines across a range of guidance scales. Figure 4-(a) plots the target reward (ImageReward) against source fidelity (LPIPS) on 100 REFL prompt images. Our approach achieves a dominant Pareto front, indicating a better editing method for any given level of editing scale.

**Performance over Different Guidance Scale.** To verify that the superior performance of our method does not simply result from using more computation, we examined the trade-off between computational efficiency and performance for both our method and the baselines. The results are summarized in Figure 4-(b); the baselines can apply additional optimization steps by increasing $N_{recur}$ and $N_{iter}$, but they still achieve lower reward and source preservation than our model at the same FLOPs. We also observe that, unlike ours, excessive guidance for the baselines leads to a reward decrease. Note that increasing $N_{iter}$ for our method does not correspond to a stronger guidance, but a better convergence toward the optimal trajectory. consequently, LPIPS does not increase with larger $N_{recur}$, but rather decreases.

**Impact of Initial Trajectory Generation Strategy.** Our main experiments use initial trajectories $\{x_t\}_{t=T}^1$ via simulating a noiseless reverse sampling path. An alternative is to generate a stochastic Markovian trajectory by applying the forward SDE process to the source image (Song et al., 2021b; Rout et al., 2025). This approach simulates a sampling path with a different noise schedule, with a fixed realization of the Brownian motion term $B_t \neq 0$. While both are effective as shown in Figure 5,

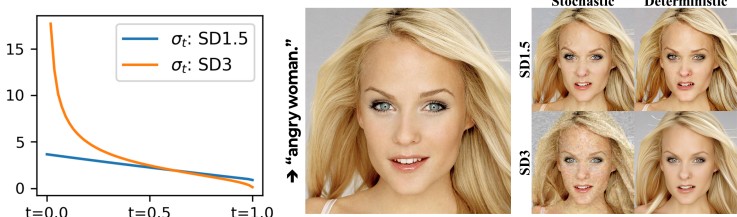

Figure 5: Selection of different initial trajectory generation strategies on different model types, with the plot of $\sigma_t$ for each model.

we found that the Markovian trajectory is more sensitive to hyperparameters and more prone to image degradation, especially in flow-matching models. This is likely because the sampling of the Markovian path has a chance to introduce an infeasible Brownian term for real-world images, where this error magnifies on flow-matching models with high $\sigma_t$. See Appendix A.1 for more detailed analyses on the choice of different initial trajectories.

## 7 CONCLUSION

In this work, we proposed a novel reward-guided image editing by formulating the task as a trajectory optimal control problem. Unlike previous guidance methods that typically rely on stepwise corrections with posterior mean, which can compromise the global structure of the image, our method treats the entire reverse diffusion trajectory as the object of optimization. Notably, our framework is training-free and broadly applicable across diffusion and flow-matching models. Our experiments across human preference optimization, style transfer, counterfactual generation, and text-guided editing demonstrate that this approach not only achieves substantial gains on reward objectives but also mitigates common pitfalls such as reward hacking and structural collapse.

### ACKNOWLEDGMENTS

This work was supported by the National Research Foundation of Korea under Grant No. RS-2024-00336454. This work was supported by the Institute for Information & communications Technology Planning & Evaluation(IITP) grant funded by the Korea government(MSIT): (RS-2019-II190075, Artificial Intelligence Graduate School Program(KAIST)) (RS-2025-02304967, AI Star Fellowship(KAIST)). This research was supported by the AI Computing Infrastructure Enhancement (GPU Rental Support) User Support Program funded by the Ministry of Science and ICT (MSIT), Republic of Korea (RQT-25-120217).

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

# A    IMPLEMENTATION DETAILS

## A.1    MORE DETAILED ALGORITHM FOR DIFFUSION AND FLOW-MATCHING MODELS

In this section, we describe more detailed processes of image editing with trajectory optimal control, as specified instances of Algorithm 1, for diffusion models (Algorithm 2) and flow-matching models (Algorithm 3), respectively.

**Simulate Trajectory.** We suggest two possible implementations of `simulate_trajectory`$(\boldsymbol{x}_1, \theta)$ for each model family to generate a plausible sampling trajectory:

- *Deterministic* trajectory for a given source image $\boldsymbol{x}_1$ can be obtained by deterministic DDIM Inversion Eq. (11) for diffusion models and time-reversed ODE Eq. (12) for flow-matching models:

$$\boldsymbol{x}_{t-dt} = \sqrt{\bar{\alpha}_{t-dt}}\left(\frac{\boldsymbol{x}_t - \sqrt{1-\bar{\alpha}_t}\boldsymbol{\epsilon}_\theta(\boldsymbol{x}_t, t)}{\sqrt{\bar{\alpha}_t}}\right) + \sqrt{1-\bar{\alpha}_{t-dt}}\boldsymbol{\epsilon}_\theta(\boldsymbol{x}_t, t) \qquad (11)$$

$$\boldsymbol{x}_{t-dt} = \boldsymbol{x}_t - \boldsymbol{v}_\theta(\boldsymbol{x}_t, t)dt. \qquad (12)$$

  These methods simulate the sampling process that leads to $\boldsymbol{x}_1$ without any stochasticity.

- *Markovian* trajectory for a given source image $\boldsymbol{x}_1$ can be obtained by simulating the forward SDE that retains the same marginal probability of $p(\boldsymbol{x}_t)$. Diffusion models can readily utilize their forward process as Eq. (13), and flow-matching models also have the corresponding forward SDE as Eq. (14) with the same marginal distribution (Rout et al., 2025) as follows:

$$\boldsymbol{x}_{t-dt} = \sqrt{\alpha_{t-dt}}\boldsymbol{x}_t + \sqrt{1-\alpha_{t-dt}}\boldsymbol{\epsilon}, \qquad \boldsymbol{\epsilon} \sim \mathcal{N}(0, \boldsymbol{I}) \qquad (13)$$

$$\boldsymbol{x}_{t-dt} = \boldsymbol{x}_t - \frac{1}{t}\boldsymbol{x}_t dt + \sqrt{\frac{2(1-t)dt}{t}}\boldsymbol{\epsilon}, \qquad \boldsymbol{\epsilon} \sim \mathcal{N}(0, \boldsymbol{I}), \qquad (14)$$

where $\alpha_{t-dt} = \frac{\bar{\alpha}_{t-dt}}{\bar{\alpha}_t}$ is the single-step noise schedule. This simulates the sampling process by Eq. (4) with $\sigma_t = \sqrt{\frac{\dot{\bar{\alpha}}_t}{\bar{\alpha}_t}}$ for diffusion models, and Eq. (5) with $\sigma_t = \sqrt{\frac{2(1-t)}{t}}$ for flow-matching models. The difference between $\hat{\boldsymbol{x}}_{t+dt|t} := \mathbb{E}[\boldsymbol{x}_{t+dt}|\boldsymbol{x}_t]$ and the simulated trajectory $\boldsymbol{x}_{t+dt}$ can be considered as the realization of the Brownian term $\boldsymbol{B}_t$,

$$\boldsymbol{B}_t := \boldsymbol{x}_{t+dt} - \left(\sqrt{\bar{\alpha}_{t+dt}}\left(\frac{\boldsymbol{x}_t - \sqrt{1-\bar{\alpha}_t}\boldsymbol{\epsilon}_\theta(\boldsymbol{x}_t, t)}{\sqrt{\bar{\alpha}_t}}\right) + \sqrt{1-\bar{\alpha}_{t+dt}-\eta_t^2}\boldsymbol{\epsilon}_\theta(\boldsymbol{x}_t, t)\right) \qquad (15)$$

$$\boldsymbol{B}_t := \boldsymbol{x}_{t+dt} - \left(\boldsymbol{x}_t + \left(2\boldsymbol{v}_\theta(\boldsymbol{x}_t, t) - \frac{1}{t}\boldsymbol{x}_t\right)dt\right), \qquad (16)$$

where $\eta_t := \sigma_t\sqrt{dt}$.

Deterministic trajectory guarantees the model will generate the source image following the obtained trajectory. On the other hand, obtaining a Markovian trajectory via noise injection requires the assumption that the source image follows the distribution learned by the pre-trained model. As the real-world image deviates from the modeled distribution, the calculated $\boldsymbol{B}_t$ becomes more infeasible as a Brownian term $\sigma_t d\boldsymbol{B}_t$. This error is often exaggerated in flow-matching models with high $\sigma_t$ (see Figure 5. Instead, multiple Markovian trajectories can be generated from a single source image, enabling diverse editing results with the same setting.

**Compute Adjoint.** In each iteration of our trajectory optimization, we compute the set of adjoint states $\{p_t\}_{t=T}^1$ using the process `compute_adjoint`$(\{\boldsymbol{x}_t\}_{t=T}^1, \theta, wr(\cdot))$, given the current trajectory, reward function $r$ and a weight parameter $w$. This is achieved by iteratively solving the partial differential equation (PDE) in Eq. (9) backward in time from $t = 1$ to $t = T$. Notably, the reward weight $w$ globally scales the magnitude of the adjoint states, thereby controlling the overall strength of the guidance applied to the trajectory.

**Update Control.** Rather than directly applying the optimal control condition $u_t = -p_t$ from Eq. (10), we employ a gradient-based update scheme. We update the control at each iteration by

---

**Algorithm 2** Image Editing via Trajectory Optimization Control with Diffusion Model

---

**Require:** Source image $\boldsymbol{x}_1$, Depth $0 < T < 1$, Number of iteration $N$, Base model $\theta$, Learning rate $\lambda$, Reward function $r(\cdot)$, Reward weight $w$, mode $\in \{$'Deterministic', 'Markovian'$\}$

1: $\eta_t = 0$ **if** mode $==$ 'Deterministic' **else** $\sqrt{\frac{1-\bar{\alpha}_{t+dt}}{1-\bar{\alpha}_t}}(1-\alpha_t)$

2: Define $\hat{\boldsymbol{x}}_{t+dt|t} := \left( \sqrt{\bar{\alpha}_{t+dt}} \left( \frac{\boldsymbol{x}_t - \sqrt{1-\bar{\alpha}_t}\boldsymbol{\epsilon}_\theta(\boldsymbol{x}_t,t)}{\sqrt{\bar{\alpha}_t}} \right) + \sqrt{1-\bar{\alpha}_{t+dt} - \eta_t^2}\boldsymbol{\epsilon}_\theta(\boldsymbol{x}_t,t) \right)$

3: **if** mode $==$ 'Deterministic' **then**

4:     $\{\boldsymbol{x}_t\}_{t=T}^1 = \texttt{DDIM\_Inversion}(x_1, \theta)$

5:     $\{\boldsymbol{B}_t\}_{t=T}^1 = \boldsymbol{0}$

6: **else**

7:     $\boldsymbol{x}_{t-dt} = \sqrt{\alpha_{t-dt}}\boldsymbol{x}_t + \sqrt{1-\alpha_{t-dt}}\epsilon$ **for** $t = 1, ..., T+dt$

8:     $\boldsymbol{B}_t = \boldsymbol{x}_{t+dt} - \hat{\boldsymbol{x}}_{t+dt|t}$ **for** $t = T, ..., 1-dt$

9: **end if**

10: $\{u_t\}_{t=T}^1 = \boldsymbol{0}$

11: **for** $iter = 1$ **to** $N$ **do**

12:     $p_1 = -w\nabla_{\boldsymbol{x}_1}r(\boldsymbol{x}_1)$

13:     $p_t = p_{t+dt} + p_{t+dt}^\top \nabla_{\boldsymbol{x}_t}(\hat{\boldsymbol{x}}_{t+dt|t} - \boldsymbol{x}_t)$ **for** $t = 1-dt, ..., T$

14:     $u_t = u_t - \lambda(u_t + p_t)$ **for** $t = 1, ..., T$

15:     $\boldsymbol{x}_{t+dt} = \hat{\boldsymbol{x}}_{t+dt|t} + u_t dt + \boldsymbol{B}_t$ **for** $t = T, ..., 1-dt$

16: **end for**

17: **return** $\boldsymbol{x}_1$

---

**Algorithm 3** Image Editing via Trajectory Optimization Control with Flow-Matching Model

---

**Require:** Source image $\boldsymbol{x}_1$, Depth $0 < T < 1$, Number of iteration $N$, Base model $\theta$, Learning rate $\lambda$, Reward function $r(\cdot)$, Reward weight $w$, mode $\in \{$'Deterministic', 'Markovian'$\}$

1: $\sigma_t = 0$ **if** mode $==$ 'Deterministic' **else** $\sqrt{\frac{2(1-t)}{t}}$

2: Define $\hat{\boldsymbol{x}}_{t+dt|t} := \boldsymbol{x}_t + \left( \boldsymbol{v}_\theta(\boldsymbol{x}_t,t) + \frac{t\sigma_t^2}{2(1-t)}\left( \boldsymbol{v}_\theta(\boldsymbol{x}_t,t) - \frac{1}{t}\boldsymbol{x}_t \right) \right)dt$

3: **if** mode $==$ 'Deterministic' **then**

4:     $\boldsymbol{x}_{t-dt} = \boldsymbol{x}_t - \boldsymbol{v}_\theta(\boldsymbol{x}_t,t)dt$ **for** $t = 1, ..., T+dt$

5:     $\{\boldsymbol{B}_t\}_{t=T}^1 = \boldsymbol{0}$

6: **else**

7:     $\boldsymbol{x}_{t-dt} = \boldsymbol{x}_t - \frac{1}{t}\boldsymbol{x}_t dt + \sqrt{\frac{2(1-t)dt}{t}}\boldsymbol{\epsilon}$ **for** $t = 1, ..., T+dt$

8:     $\boldsymbol{B}_t = \boldsymbol{x}_{t+dt} - \hat{\boldsymbol{x}}_{t+dt|t}$ **for** $t = T, ..., 1-dt$

9: **end if**

10: $\{u_t\}_{t=T}^1 = \boldsymbol{0}$

11: **for** $iter = 1$ **to** $N$ **do**

12:     $p_1 = -w\nabla_{\boldsymbol{x}_1}r(\boldsymbol{x}_1)$

13:     $p_t = p_{t+dt} + p_{t+dt}^\top \nabla_{\boldsymbol{x}_t}(\hat{\boldsymbol{x}}_{t+dt|t} - \boldsymbol{x}_t)$ **for** $t = 1-dt, ..., T$

14:     $u_t = u_t - \lambda(u_t + p_t)$ **for** $t = 1, ..., T$

15:     $\boldsymbol{x}_{t+dt} = \hat{\boldsymbol{x}}_{t+dt|t} + u_t dt + \boldsymbol{B}_t$ **for** $t = T, ..., 1-dt$

16: **end for**

17: **return** $\boldsymbol{x}_1$

---

taking a gradient step with learning rate $\lambda$ to minimize $L_2$ distance $\|u_t + p_t\|_2^2$. Note that while we describe the most naive gradient ascent in Algorithm 2 and Algorithm 3, more advanced optimizers can also be utilized for more stable optimization. Empirically, we find that even a single optimization step per iteration is sufficient to achieve stable optimization while maintaining alignment with the PMP conditions.

## A.2 HYPERPARAMETER SELECTION

Table 6 lists the detailed hyperparameters for our method and baselines on different image editing scenarios. The hyperparameter notation for the guided sampling baselines follows TFG (Ye et al.,

| StableDiffusion 1.5 (Image Resolution: $512 \times 512$) | | | | |
|---|---|---|---|---|
| Method | Gradient Ascent | Inversion + DPS | Inversion + FreeDoM | Inversion + TFG | Ours |
| Human Preference | N=100, $\lambda$=2.0 | Inversion depth=0.7, $\rho_t = 3.0$ | Inversion depth=0.7, $N_{recur} = 2, \rho_t = 1.0$ | Inversion depth=0.7, $N_{recur} = 1, \rho_t = 1.0$, $N_{iter} = 4, \mu_t = 0.5$, $\bar{\gamma} = 0.1$ | $T = 0.5, N = 20$, $w = 500$ |
| Style Transfer | N=100, $\lambda$=3.0 | Inversion depth=0.7, $\rho_t = 15.0$ | Inversion depth=0.7, $N_{recur} = 2, \rho_t = 7.5$ | Inversion depth=0.7, $N_{recur} = 1, \rho_t = 10.0$, $N_{iter} = 4, \mu_t = 1.0$, $\bar{\gamma} = 0.1$ | $T = 0.5, N = 20$, $w = 200$ |
| Counterfactual Generation | N=100, $\lambda$=1.0 | Inversion depth=0.7, $\rho_t = 1.0$ | Inversion depth=0.7, $N_{recur} = 2, \rho_t = 0.4$ | Inversion depth=0.7, $N_{recur} = 1, \rho_t = 1.0$, $N_{iter} = 4, \mu_t = 0.1$, $\bar{\gamma} = 0.1$ | $T = 0.5, N = 20$, $w = 50$ |
| Text-Guided Image Editing | N=100, $\lambda$=1.5 | Inversion depth=0.7, $\rho_t = 40.0$ | Inversion depth=0.7, $N_{recur} = 2, \rho_t = 20.0$ | Inversion depth=0.7, $N_{recur} = 1, \rho_t = 30.0$, $N_{iter} = 4, \mu_t = 2.5$, $\bar{\gamma} = 0.1$ | $T = 0.5, N = 20$, $w = 1000$ |
| StableDiffusion 3 (Image Resolution: $768 \times 768$) | | | | |
| Method | Gradient Ascent | Inversion + DPS | Inversion + FreeDoM | Inversion + TFG | Ours |
| Human Preference | N=100, $\lambda$=2.0 | Inversion depth=0.7, $\rho_t = 5.0$ | Inversion depth=0.7, $N_{recur} = 2, \rho_t = 5.0$ | Inversion depth=0.7, $N_{recur} = 1, \rho_t = 5.0$, $N_{iter} = 4, \mu_t = 1.0$, $\bar{\gamma} = 0.1$ | $T = 0.7, N = 15$, $w = 500$ |
| Style Transfer | N=100, $\lambda$=3.0 | Inversion depth=0.7, $\rho_t = 50.0$ | Inversion depth=0.7, $N_{recur} = 2, \rho_t = 20$ | Inversion depth=0.7, $N_{recur} = 1, \rho_t = 40.0$, $N_{iter} = 4, \mu_t = 2.5$, $\bar{\gamma} = 0.1$ | $T = 0.7, N = 15$, $w = 1000$ |
| Counterfactual Generation | N=100, $\lambda$=1.0 | Inversion depth=0.7, $\rho_t = 7.0$ | Inversion depth=0.7, $N_{recur} = 2, \rho_t = 3.0$ | Inversion depth=0.7, $N_{recur} = 1, \rho_t = 7.0$, $N_{iter} = 4, \mu_t = 0.5$, $\bar{\gamma} = 0.1$ | $T = 0.7, N = 15$, $w = 200$ |
| Text-Guided Image Editing | N=100, $\lambda$=1.5 | Inversion depth=0.7, $\rho_t = 75.0$ | Inversion depth=0.7, $N_{recur} = 2, \rho_t = 30.0$ | Inversion depth=0.7, $N_{recur} = 1, \rho_t = 60.0$, $N_{iter} = 4, \mu_t = 2.5$, $\bar{\gamma} = 0.1$ | $T = 0.7, N = 15$, $w = 1000$ |

Table 6: Hyperparameter settings for the quantitative results in Section 5.2 on different base models, methods, and experiment scenarios.

2024). Inversion depth refers to the noise level, the same as $T$ in ours. $\rho_t$ and $\mu_t$ denote the guidance strength multiplied by the $\nabla_{\boldsymbol{x}_t} r(\hat{\boldsymbol{x}}_{1|t})$ and $\nabla_{\hat{\boldsymbol{x}}_{1|t}} r(\hat{\boldsymbol{x}}_{1|t})$, respectively, where $\hat{\boldsymbol{x}}_{1|t}$ denotes the posterior mean. $N_{iter}$ represents the number of guidance updates performed in a single timestep, and $N_{recur}$ is the number of times the same timestep is repeated with a forward noise injection. $\bar{\gamma}$ is the noise scale injected to $\boldsymbol{x}_{1|t}$ for TFG, which is fixed at 0.1.

**Robustness to Hyperparameters.** We analyzed the impact of key hyperparameter selection in our algorithm in Algorithm 2, namely the inversion depth $T$ and the number of optimization iterations $N$. As shown in Figure 6, the inversion depth $T$ controls the trade-off between editing strength and source consistency, aligning with observations in previous image editing literature. When $T \rightarrow 1$ (*i.e.*, shallow noise), the editing effect is minimal as the trajectory has little room to deviate. As $T \rightarrow 0$ (*i.e.*, pure noise), the potential for editing increases, but at the risk of losing fidelity to the source image. Crucially, the number of iterations $N$ governs the convergence of the output trajectory rather than the guidance strength. As illustrated in Figure 6, the final result is not highly sensitive to $N$ provided that $N$ is sufficient for convergence. However, omitting the iterative process entirely ($N = 1$ with $\lambda = 1.0$) leads to significant artifacts. This is because the control computed from the initial trajectory is no longer optimal for the modified states. Our iterative refinement is therefore essential to ensure the trajectory converges to produce high-reward images.

To further investigate stability, we visualize the behavior of our method and baselines under increasing guidance scales (*i.e.*, reward weight $w$ for ours, and $\rho_t, \mu_t$ for baselines) in Figure 7. While the quantitative trade-off was shown in Figure 4-(a), these qualitative results highlight a distinct difference in robustness. As the guidance scale increases, baselines begin to exhibit severe degradation, including color saturation, artifacts, and structural corruption. In contrast, our method achieves

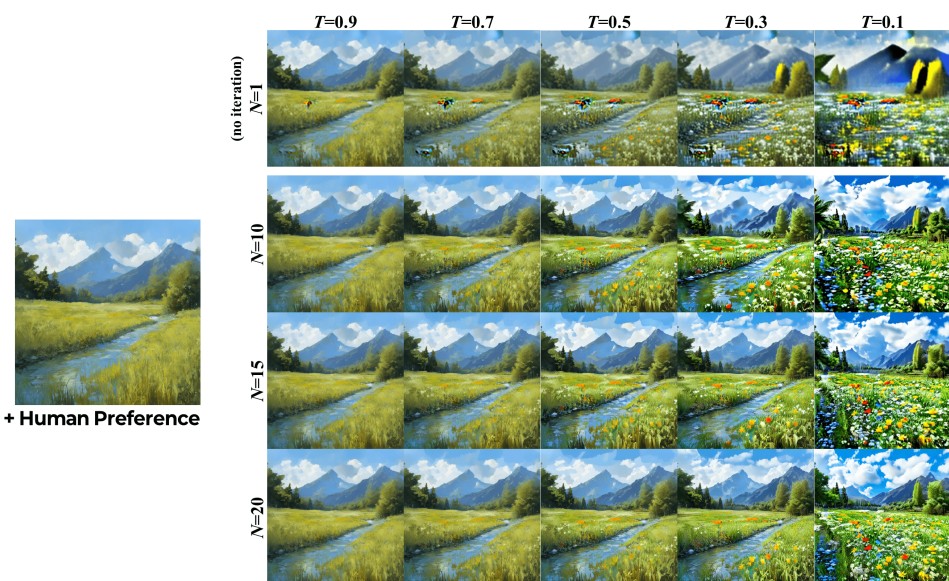

Figure 6: Qualitative ablation study on different choices of hyperparameters for the depth $T$ and the number of iterations $N$. The text prompt for the alignment is "*colorful painting, river flowing grass field with flowers.*".

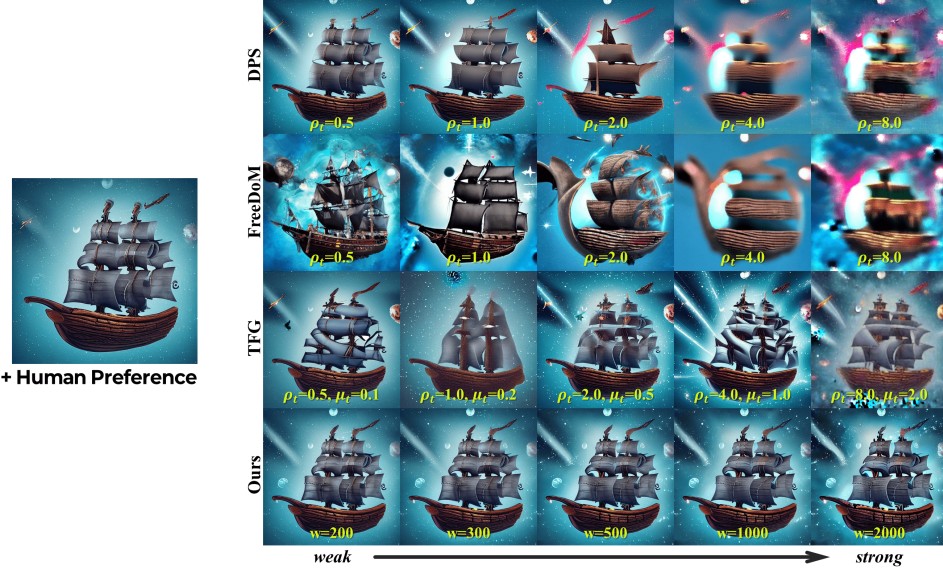

Figure 7: Qualitative results on varying guidance scale. The other hyperparameters except $\{\rho_t, \mu_t, w\}$ follow the configuration in Table 6. The text prompt for the alignment is "*pirate ship, flowing through cosmic nebula.*".

significantly higher target rewards while demonstrating a smooth and progressive emphasis on the objective, without compromising image quality.

## A.3 USER STUDY

We conducted a human-subject study with 42 participants from the general population recruited via an online platform. As shown in Figure 8, each participant was presented with a series of source–edited image pairs and asked to rate the edited images along three criteria:

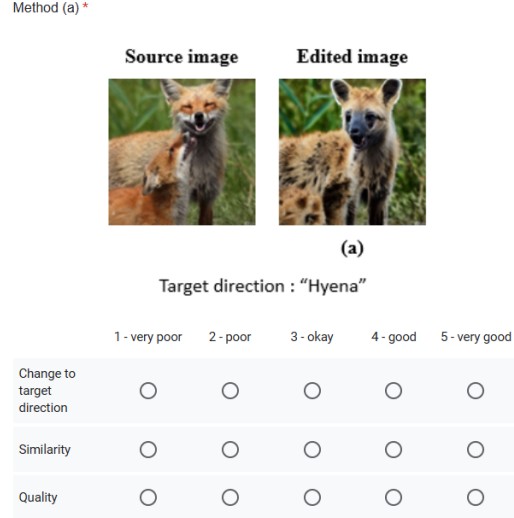

Figure 8: An example question from our user study survey.

1. **Change**: Does the edited image show meaningful and noticeable changes in the target direction (e.g., category shift, text description, or style transfer)?

2. **Similarity**: Does the edited image remain faithful to the source image, including background and other non-target regions?

3. **Quality**: Does the edited image look realistic without obvious artifacts or distortions?

They were instructed to rate each image on a 5-point Likert scale (1 = very poor, 5 = very good). Table 5 summarizes the user ratings across the three criteria. These results confirm that our model produces edits that are not only aligned with the intended modifications but also visually convincing and coherent.

## B ADDITIONAL RESULTS

### B.1 RESULTS ON FLOW-MATCHING MODELS

While the main manuscript focuses on the performance of our methods on the diffusion-based models, this section presents qualitative results on a state-of-the-art flow-matching model, StableDiffusion 3, to validate the generality of our method. The experimental protocol remains identical to the experiments in the main paper. Note that all of the baseline methods were originally suggested for diffusion models, and we re-implemented their calculation of the posterior mean and forward noise process analogous to flow-matching models. As shown in Table 7, our method maintains its superior performance on the flow-matching model, exhibiting consistent behavior across both model families.

### B.2 CONNECTION BETWEEN OPTIMAL CONTROL TERM AND GUIDED SAMPLING

In this section, we discuss how the suggested method can be related to the guided sampling methods. In the diffusion model sampling process with the noisy sample $\hat{x}_t$, DPS and many of the suggested guided sampling variations (Chung et al., 2023; Yu et al., 2023; He et al., 2023; Ye et al., 2024) calculate the gradient of the objective function at the posterior mean $\hat{x}_{1|t}$ with respect to $x_t$, and this guidance term is added into the denoising direction.

Here, we show that this guidance term suggested in DPS $\nabla_{x_t} r(\hat{x}_{1|t})$ can be explained from a perspective of the solution of the optimal control problem:

**Human Preference**

| Method | Target reward ImageReward[↑] | Validation metrics HPSv2[↑] | CLIPScore[↑] | Aesthetic[↑] | Source preservation LPIPS[↓] | CLIP-I$_{src}$[↑] |
|---|---|---|---|---|---|---|
| None | 0.1542 | 0.2385 | 0.2887 | 6.0516 | 0.0000 | 1.0000 |
| Gradient Ascent | **1.9088** | 0.2247 | 0.2877 | 5.5775 | **0.1474** | **0.9195** |
| Inversion+DPS | 1.4169 | 0.2189 | 0.2552 | 5.7227 | 0.3767 | 0.7896 |
| Inversion+FreeDoM | 1.5887 | 0.2288 | 0.2305 | 5.7446 | 0.5460 | 0.6893 |
| Inversion+TFG | 1.5162 | 0.2216 | 0.2745 | 5.6072 | 0.3083 | 0.8537 |
| Ours | 1.8529 | **0.2400** | **0.2890** | **6.1730** | 0.2475 | 0.9013 |

**Style Transfer**

| Method | Target reward $||\Delta G||_F$[↓] | Validation metrics CLIP-I$_{sty}$ [↑] | DINO$_{sty}$[↑] | Source preservation CLIP-I$_{src}$[↑] |
|---|---|---|---|---|
| None | 12.190 | 0.4757 | 0.1236 | 1.0000 |
| Gradient Ascent | 4.8742 | 0.5270 | 0.1953 | **0.8374** |
| Inversion+DPS | 5.3983 | 0.5553 | 0.1774 | 0.6617 |
| Inversion+FreeDoM | 4.9643 | 0.5466 | 0.2091 | 0.6365 |
| Inversion+TFG | 5.4176 | 0.5495 | 0.1922 | 0.6758 |
| Ours | **4.5333** | **0.5633** | **0.2201** | 0.7666 |

**Counterfactual Generation**

| Method | Target reward Logit$_{tgt}$[↑] | Validation metrics CLIPScore [↑] | Source preservation LPIPS[↓] | CLIP-I$_{src}$[↑] |
|---|---|---|---|---|
| None | 4.8722 | 0.1452 | 0.0000 | 1.0000 |
| Gradient Ascent | **24.875** | 0.1908 | **0.2246** | **0.8203** |
| Inversion+DPS | 21.628 | 0.1874 | 0.3852 | 0.6498 |
| Inversion+FreeDoM | 23.085 | 0.1984 | 0.4017 | 0.6241 |
| Inversion+TFG | 21.538 | 0.1872 | 0.3846 | 0.6506 |
| Ours | 24.572 | **0.2044** | 0.2743 | 0.7040 |

**Text-guided Image Editing**

| Method | Target reward CLIPScore[↑] | Validation metrics ImageReward[↑] | HPSv2 [↑] | Source preservation LPIPS[↓] | CLIP-I$_{src}$[↑] |
|---|---|---|---|---|---|
| None | 0.1760 | -0.2404 | 0.2233 | 0.0000 | 1.0000 |
| Gradient Ascent | **0.3567** | -0.2331 | 0.2193 | **0.1250** | **0.6660** |
| Inversion+DPS | 0.2915 | -0.4124 | 0.2106 | 0.3262 | 0.5665 |
| Inversion+FreeDoM | 0.3060 | -0.2091 | 0.2242 | 0.4571 | 0.5281 |
| Inversion+TFG | 0.2944 | -0.3118 | 0.2093 | 0.3386 | 0.5597 |
| Ours | 0.3491 | **-0.1308** | **0.2272** | 0.2439 | 0.6011 |

Table 7: Quantitative performance of the proposed method and baselines with StableDiffusion 3. **Bold**: best, underline: second best.

**Proposition 1.** *The guidance term by DPS is equivalent to the negative adjoint state $-p_t$ under the optimal control problem in Eq. (7), calculated with a one-step sampling trajectory from $\boldsymbol{x}_t$.*

*proof.* Note that a one-step sampling from $\boldsymbol{x}_t$ to a clean image domain(*e.g.*, $t = 1$) gives $\hat{\boldsymbol{x}}_{1|t}$ as a terminal point. When the adjoint state $p_t$ is calculated in this one-step trajectory according to Eq. (9), we get

$$p_0 = -\nabla_{\hat{\boldsymbol{x}}_{1|t}}(wr(\hat{\boldsymbol{x}}_{1|t})), \tag{17}$$

$$p_t = p_0 + \nabla_{\boldsymbol{x}_t}[(\hat{\boldsymbol{x}}_{1|t} - \boldsymbol{x}_t)^\top p_0] \tag{18}$$

$$= (I + J_{\boldsymbol{x}_t}(\hat{\boldsymbol{x}}_{1|t})^\top - I)p_0 \tag{19}$$

$$= J_{\boldsymbol{x}_t}(\hat{\boldsymbol{x}}_{1|t})^\top p_0 \tag{20}$$

where $J_{\boldsymbol{x}_t}(\hat{\boldsymbol{x}}_{1|t})$ denotes a Jacobian matrix, defined as $J_{ij} = \frac{\partial \hat{\boldsymbol{x}}_{1|t}^i}{\partial \boldsymbol{x}^j}$, where $\boldsymbol{x}^i$ is $i$-th element of $\boldsymbol{x}$. When we put Eq. (17) to Eq. (20), from the chain rule,

| | | Gradient Ascent | Inversion +DPS | Inversion +FreeDoM | Inversion +TFG | Ours |
|---|---|---|---|---|---|---|
| Required time [s/image] | StableDiffusion 1.5 | 23.74 | 14.77 | 23.09 | 37.26 | 60.63 |
| | StableDiffusion 3 | 27.55 | 13.26 | 20.31 | 30.50 | 41.97 |
| FLOPs [$10^{12}$/image] | StableDiffusion 1.5 | 277.91 | 155.93 | 284.59 | 543.47 | 590.93 |
| | StableDiffusion 3 | 592.51 | 353.95 | 676.09 | 863.84 | 1950.26 |

Table 8: Required time and FLOPs for each method with different base models. The hyperparameters in the Human Preference row of Table 6 with the ImageReward reward function were used. We ran our experiments with StableDiffusion 3 in half-precision floating point format(float16).

$$p_t = -J_{\boldsymbol{x}_t}(\hat{\boldsymbol{x}}_{1|t})^\top \nabla_{\hat{\boldsymbol{x}}_{1|t}}(wr(\hat{\boldsymbol{x}}_{1|t})) \tag{21}$$

$$= -w\nabla_{\boldsymbol{x}_t}r(\hat{\boldsymbol{x}}_{1|t}), \tag{22}$$

where Eq. (22) is equivalent to the guidance term utilized by DPS with a sign reversed. □

This perspective of previous guided sampling methods emphasizes the advantage of our method; it utilizes a multi-step trajectory that ends with a fully detailed source image endpoint. It also iteratively refines the control term to balance the optimization and the guidance term regularization, where previous guidance terms cannot provide a theoretically appropriate guidance strength.

## C    DISCUSSION ON RELATED FLOW-BASED EDITING METHODS

Recent works have explored the steering and editing of Rectified Flow (ReFlow) models, which share a conceptual motivation with our control-based approach. We first summarize the related works and discuss to clarify the distinct contributions of our paper.

**RF-Solver** (Wang et al., 2024): RF-solver is proposed to reduce inversion and reconstruction errors using a higher-order ODE sampler based on Taylor expansion. Then, RF-Edit is used for editing by storing and sharing self-attention features from the inversion path to the editing path.

**FireFlow** (Deng et al., 2024): FireFlow addresses the computational cost of high-order solvers by introducing an efficient second-order solver that reuses stored mid-point velocities calculated from the previous step.

**FlowChef** (Patel et al., 2024): FlowChef proposes an inversion-free framework for steering ReFlow models. It applies inference-time guidance by optimizing the trajectory at each step. This is achieved by estimating the final output $\hat{x}_0$ and obtaining the gradient of the loss functions (e.g., a mask-based L2 loss or classifier loss) to update the current state $x_t$.

Our work differs from prior approaches in two key aspects. Unlike existing methods, which focus on text-prompt–based editing within the ReFlow family, our framework addresses the more general setting of reward-guided editing without text conditioning and can incorporate any differentiable reward signal (e.g., human preference scores, aesthetic models, classifier logits). Methodologically, our framework formulates editing as a trajectory optimal control problem: starting from an initial inversion, we optimize the entire generation path by solving adjoint-state equations from PMP to update a time-varying control. This yields a trajectory-level optimization that does not rely on attention modulation or user-provided masks to edit images.

## D    LIMITATIONS AND FUTURE WORK

Our framework, grounded in trajectory optimal control, has several inherent limitations: First, it fundamentally requires the reward function to be differentiable, as the computation of the adjoint state relies on its gradient. While this assumption is shared for most guided sampling methods, this prerequisite limits our method to directly applying to objectives that are non-differentiable or discrete, such as direct human feedback. However, this limitation represents a promising avenue for future work;

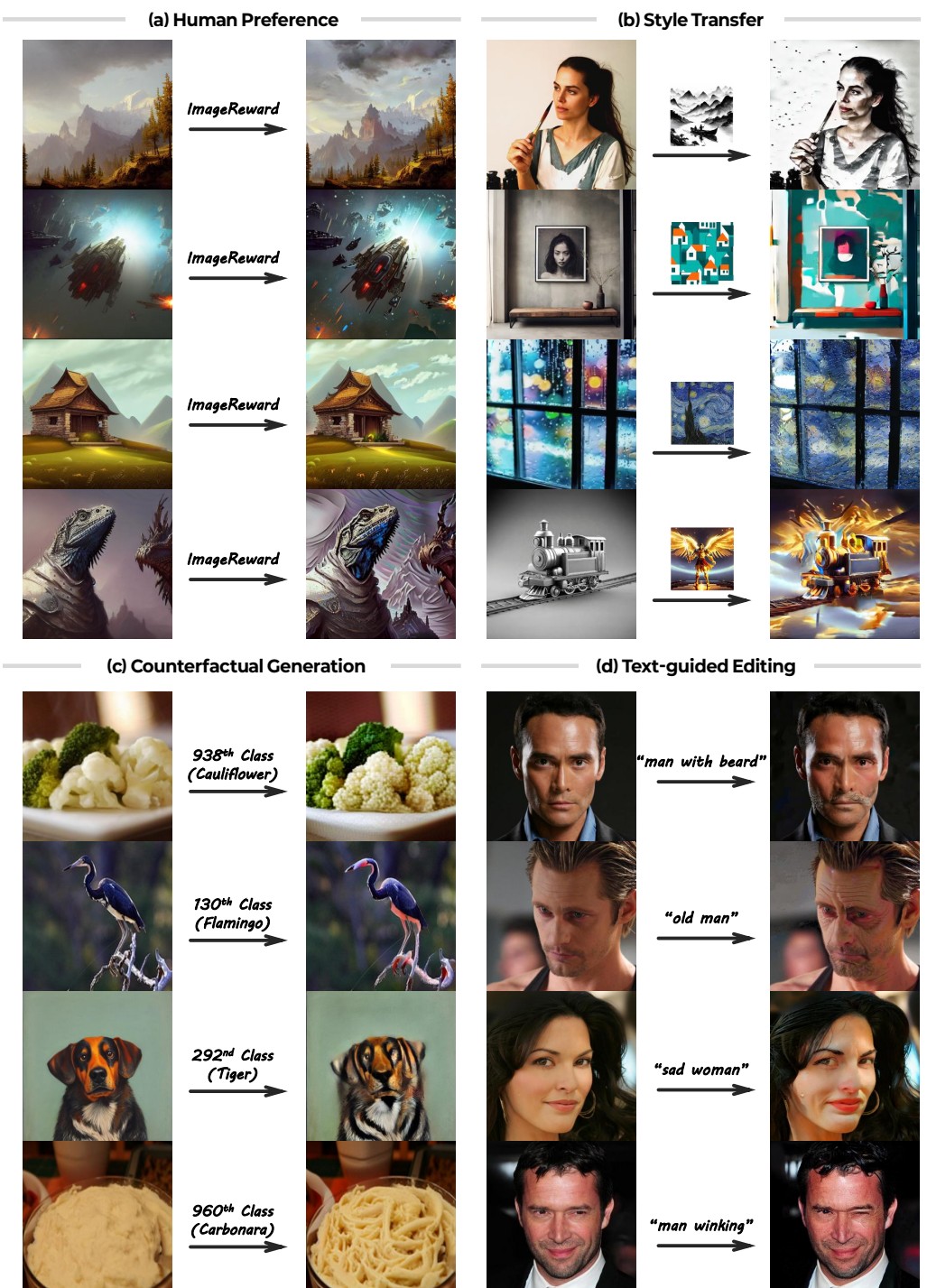

Figure 9: Additional qualitative image editing examples of our method and source images.

the framework could be extended to black-box settings by employing Zeroth-Order gradient estimation techniques to approximate the reward gradients numerically (Kim et al., 2025a). Second, the number of model evaluations in our method is proportional to $T$ and $N$, leading to about $40 \sim 60\%$ more required time than guided sampling-based methods, as shown in Table 8. Nevertheless, as demonstrated in our efficiency-performance trade-off analysis (Figure 4), our method establishes a superior Pareto frontier compared to baselines. This indicates that the performance gain justifies

the additional computational cost, and our method maintains its advantage even when baselines are given an equivalent computational budget. Finally, while this work comprehensively validates the framework for 2D image editing, its generalization to other domains like video, 3D models, or audio remains for future research. Additionally, investigating optimal reward-aware trajectory initialization strategies beyond deterministic inversion could be valuable for further accelerating convergence.

## E  THE USE OF LARGE LANGUAGE MODELS (LLMs)

LLMs were not involved in research ideation or methodological design and were only used for the purpose of minor expression refinement. The authors retain full responsibility for all scientific content.

