# OpenReview forum: "Training-Free Reward-Guided Image Editing via Trajectory Optimal Control"
_ICLR.cc/2026/Conference — ICLR 2026 Poster_

### Official Review · Reviewer_HoKy · 2025-10-27

**Soundness:** 2
**Presentation:** 2
**Contribution:** 2
**Rating:** 4
**Confidence:** 5

**Summary:**

This paper introduces a novel, training-free framework for reward-guided image editing applicable to both diffusion and flow-matching models. The core innovation is formulating the editing process as a trajectory optimal control problem. This approach treats the reverse diffusion process, starting from a source image, as a controllable trajectory. By leveraging the principles of Pontryagin's Maximum Principle (PMP), the authors develop an iterative adjoint-state optimization algorithm that steers the entire trajectory toward a terminal state that maximizes an arbitrary differentiable reward function, while explicitly minimizing the control effort to maintain fidelity to the source image structure.

The method is validated across four diverse editing tasks: Human Preference Optimization, Style Transfer, Counterfactual Generation, and Text-guided Editing. Experiments show that this trajectory-level optimization approach achieves a superior balance between reward maximization and source image fidelity compared to existing inversion-based guided sampling techniques, effectively mitigating issues like "reward hacking" and structural degradation.

**Strengths:**

Principled Formulation based on Optimal Control (OC): The paper provides a strong theoretical foundation by framing image editing as a trajectory optimal control problem and deriving the iterative update rules using PMP. This approach is superior to heuristic, step-wise guidance methods as it optimizes the entire path, leading to better structural coherence and a more theoretically grounded way to balance editing (reward) and fidelity (control regularization).

Broad Applicability (Generality): The framework is designed to be training-free and is unified under the Stochastic Differential Equation (SDE) framework, making it applicable to both Diffusion Models (SD 1.5) and Flow-Matching Models (SD 3). This generality significantly broadens its impact and utility.

Superior Fidelity-Reward Trade-off: Quantitatively and qualitatively, the method consistently establishes a dominant Pareto front between reward alignment and source fidelity (Figure 4a). In contrast to baselines which often suffer from structural collapse or reward hacking, this method achieves high rewards while maintaining high LPIPS and $CLIP-I_{src}$ (Tables 1, 2, 3, 4).

Mitigation of Common Failure Modes: The trajectory optimization successfully avoids common failure modes of guided sampling methods, such as reward hacking (where GA introduces severe artifacts) and structural degradation (where stepwise guidance corrupts high-frequency details). The user study (Table 5) confirms the superior perceptual quality and faithfulness of the generated edits.

**Weaknesses:**

Fundamental Dependence on Differentiable Rewards (Critical Limitation): The entire framework hinges on the reward function $r(\cdot)$ being differentiable, as the adjoint state calculation explicitly requires $\nabla_{x_1} r(x_1)$. This strictly limits its application to objectives that are inherently smooth or can be approximated by differentiable proxies, excluding direct application of discrete or non-differentiable objectives (like direct human feedback ratings). This restricts its practical use in true preference learning settings.

High Computational Cost and Efficiency Concerns: The method is computationally expensive. As shown in Table 8, the required time for the optimization is substantially higher (approximately 80% more time than guided sampling baselines) because the iterative process necessitates multiple, full, backwards and forwards passes over the model's trajectory (proportional to $T$ and $N$). While the results are better, the practical deployment cost is a major constraint.

Sensitivity to Optimization Hyperparameters: The method introduces new critical hyperparameters, notably the inversion depth $T$ (where the optimization starts) and the number of iterations $N$. As shown in Figure 5, the performance is highly sensitive to $T$ (controlling the fidelity-edit trade-off) and $N$ (critical for convergence and stability). The need for careful tuning of $T$ and $N$ for every new task or model reduces the "training-free" simplicity.

Simplistic Trajectory Initialization: The paper primarily relies on deterministic DDIM/ODE inversion for the initial trajectory. While the effects of Markovian (stochastic) initialization are discussed, the deterministic approach, while simple, may place the starting trajectory on a rigid path that is far from the optimal reward direction, potentially leading to slow convergence in complex editing scenarios. Further investigation into optimal reward-aware initialization could be beneficial.

**Questions:**

above

---

> ### Author Response · Authors · 2025-11-22
> **Rebuttal from Authors**
>
> We thank Reviewer ``HoKy`` for the constructive feedback. Below, we provide point-by-point responses on your questions and concerns.
>
>
> - **W1: Fundamental Dependence on Differentiable Rewards:** The entire framework hinges on the reward function r() being differentiable, as the adjoint state calculation explicitly requires $\nabla_{x_1}r(x_1)$. This strictly limits its application to objectives that are inherently smooth or can be approximated by differentiable proxies, excluding direct application of discrete or non-differentiable objectives (like direct human feedback ratings). This restricts its practical use in true preference learning settings.
>
> We acknowledge the reviewer's valid point regarding the dependence on differentiable rewards. We first note that this constraint is inherent to the class of gradient-based guidance methods (e.g., DPS, FreeDoM, TFG) that operate without model fine-tuning, as discussed in Limitation Section. Our primary contribution focuses on maximizing the efficacy of these available gradients through a trajectory optimal control formulation. We think that considering scenarios where the reward gradient is unavailable lies in a different stream of research.
>
> That said, this does not strictly limit the use of non-differentiable objectives. For example, our framework can be extended to such settings by employing Zeroth-Order gradient estimation techniques [1, 2]. By evaluating the non-differentiable reward on a set of locally perturbed samples around the terminal image $x_1$, we can derive a numerical approximation of the gradient. This estimated gradient can directly replace the analytic gradient $\nabla_{x_1} r(x_1)$ in the terminal adjoint condition, allowing the remainder of our iterative optimization to proceed unchanged. We have incorporated a detailed discussion of this extensibility in our future work section of the revised version.
>
> - **W2: High Computational Cost and Efficiency Concerns:** The method is computationally expensive. As shown in Table 8, the required time for the optimization is substantially higher (approximately 80% more time than guided sampling baselines) because the iterative process necessitates multiple, full, backwards and forwards passes over the model's trajectory (proportional to T and N). While the results are better, the practical deployment cost is a major constraint.
>
> We acknowledge that our method incurs higher computational costs, specifically approximately 40% more FLOPs and 80% longer inference time compared to TFG, as noted in our revised Table 8. However, we emphasize that our computational complexity scales linearly with hyperparameters, similar to inversion-based baselines.
>
> To address the concern regarding practical utility, we present a computational efficiency-performance trade-off analysis in Figure 4-(b) of the revised version. This Pareto frontier demonstrates that the higher cost is justified by a superior performance ceiling. Even when baselines are allocated an equivalent computational budget by increasing refinement steps ($N_{recur}, N_{iter}$), they consistently achieve lower reward and source fidelity than our method. Furthermore, we observed that excessive guidance in baselines often leads to performance degradation, whereas our method utilizes the additional computation to stably converge toward the optimal trajectory without such deterioration.

---

> ### Author Response · Authors · 2025-11-22
> **Rebuttal from Authors (continue)**
>
> - **W3: Sensitivity to Optimization Hyperparameters:** The method introduces new critical hyperparameters, notably the inversion depth T(where the optimization starts) and the number of iterations N. As shown in Figure 5, the performance is highly sensitive to T(controlling the fidelity-edit trade-off) and N (critical for convergence and stability). The need for careful tuning of T and N for every new task or model reduces the "training-free" simplicity.
>
> We clarify that our framework effectively reduces hyperparameter complexity compared to state-of-the-art baselines. For instance, TFG requires tuning up to six hyperparameters (including inversion step $T$, $N_{iter}, N_{recur}, \rho_t, \mu_t, \bar{\gamma}$), whereas our method primarily relies on the inversion depth $T$, number of iteration $N$ and reward weight $w$.
>
> As visualized in the revised Figure 6, once $N$ is sufficient for convergence, the output images remain consistent across different $N$ values. Therefore, as discussed in Appendix A.2, $N$ does not requires careful tuning since it governs trajectory convergence rather than guidance strength. Furthermore, the inversion depth $T$ acts as an intuitive control for the trade-off between editability and source fidelity. Since $T$ shows a monotonic correlation with fidelity, we emphasize that this represents user controllability rather than unstable sensitivity.
>
> - **W4: Simplistic Trajectory Initialization:** The paper primarily relies on deterministic DDIM/ODE inversion for the initial trajectory. While the effects of Markovian (stochastic) initialization are discussed, the deterministic approach, while simple, may place the starting trajectory on a rigid path that is far from the optimal reward direction, potentially leading to slow convergence in complex editing scenarios. Further investigation into optimal reward-aware initialization could be beneficial.
>
> We appreciate the reviewer's insightful suggestion regarding trajectory initialization. We first emphasize that under our problem setting of “optimization of the unmodified base model trajectory”, the initial trajectory needs to be something that is sampled from our “base diffusion model”. As discussed in the Discussion Section and Appendix A.1 of the manuscript, we have explicitly investigated the impact of Markovian (stochastic) initialization compared to the deterministic approach.
>
> While stochastic initialization allows for broader exploration, our empirical results in Figure 5 indicate that it is more sensitive to hyperparameters and prone to structural degradation, particularly in flow-matching models. This is likely because the noise injection in stochastic paths can introduce deviated samples which are infeasible for real-world images and are not perfectly aligned with the model's prior. This deviation can be magnified during reverse sampling in flow-matching models, where the diffusion coefficient $\sigma_t$ in Eq. 3 is typically larger than that of diffusion models. Therefore, we opted for deterministic inversion as a more robust and structurally faithful starting point, ensuring that the subsequent trajectory optimization focuses on reward alignment rather than correcting inversion errors. We agree that exploring optimal reward-aware initialization is a promising avenue for future work to further accelerate convergence.
>
> ---
>
>
> [1] "A primer on zeroth-order optimization in signal processing and machine learning: Principals, recent advances, and applications." IEEE Signal Processing Magazine, 2020
>
> [2] "Random gradient-free minimization of convex functions." Foundations of Computational Mathematics, 2017

---

### Official Review · Reviewer_DZiS · 2025-10-29

**Soundness:** 3
**Presentation:** 3
**Contribution:** 3
**Rating:** 6
**Confidence:** 3

**Summary:**

This paper introduces a training-free framework for reward-guided image editing using trajectory optimal control. The authors regard the image editing procedure as the optimal control of a reverse sampling trajectory from a diffusion or flow-matching generative model. Leveraging Pontryagin’s Maximum Principle, the method iteratively updates adjoint states and control terms to steer the generation trajectory, producing edited images that balance reward maximization with fidelity to the source image. Experiments on various tasks such as human preference optimization, style transfer, counterfactual generation, and text-guided editing demonstrate the superior performance compared to baselines.

**Strengths:**

- Treating the reverse diffusion process as a controllable trajectory and solve the control problem by using Pontryagin’s Maximum Principle is novel to my knowledge.
- The derivation and mathematical formulation is solid and intuitive.
- The method is training-free, and can be applied on various base models such as SD1.5 and SD3.
- The paper is clear and well-written.

**Weaknesses:**

- As stated in the paper, the proposed method requires the reward function to be differentiable and the gradient need to be calculated. Would this cause notable extra computational overhead? Authors are expected to provide the comparison of inference time between the proposed method and baselines.
- Generally, I think the experiments are insufficient, which are still not enough to illustrate the effectiveness of the proposed method:
    1. Authors are expected to provide the results on FLUX, which is one of the most powerful image generation foundation models.
    2. Missing baselines. Currently, there emerges a number of image editing methods based on flow-matching-based models such as [1,2,3], authors should compare the image editing performance between the proposed method and these baselines.
    3. The qualitative results are not enough and some flaws still exists (for example, in second row in Figure 3, the ears of the cat are not clear in the image produced by the proposed method). Authors are expected to provide more comprehensive qualitative results, which are important to evaluate the performance.

**Questions:**

See weakness.

---

> ### Author Response · Authors · 2025-11-22
> **Rebuttal from Authors**
>
> We appreciate the reviewer ``DZiS`` for your detailed comments and valuable suggestions. Below, we provide thorough point-by-point responses to address your concerns.
>
> - **W1:** As stated in the paper, the proposed method requires the reward function to be differentiable and the gradient need to be calculated. Would this cause notable extra computational overhead? Authors are expected to provide the comparison of inference time between the proposed method and baselines.
>
> Regarding the computational overhead, we first clarify that calculating the gradient of the reward function is a standard requirement shared by all optimization-based baselines (DPS, FreeDoM, TFG). Consequently, the peak VRAM usage remains the same across methods ($\sim$14.5GB) using Stable Diffusion 1.5.
>
> Furthermore, we have provided a detailed comparison of inference time and FLOPs in Table 8 of the revised manuscript. We acknowledge that our method incurs approximately 40% more FLOPs and 80% longer inference time than TFG. However, as demonstrated in the FLOPs-performance trade-off analysis in Figure 4-(b), this additional cost is justified: even when baselines utilize increased computational budgets to match our FLOPs, they fail to achieve comparable reward maximization and source fidelity.
>
> - **W2:** Authors are expected to provide the results on FLUX, which is one of the most powerful image generation foundation models.
>
> We appreciate the reviewer’s suggestion to validate our method on the state-of-the-art FLUX model. Since FLUX is based on the flow-matching formulation, our proposed framework is directly applicable without modification. We have conducted the human preference optimization task with FLUX and included the full quantitative results below:
>
> | Method | [target]ImageReward | [valid.]HPSv2 | [valid.]CLIP | [valid.]Aesthetic | [source]LPIPS | [source]CLIP-I$_{src}$ |
> | -- | :--: | :--: | :--: | :--: | :--: | :--: |
> | None | 0.1542 | 0.2385 | 0.2887 | 6.0516 | 0.0000 | 1.0000 |
> | GA | **1.9088** | 0.2247 | $\underline{0.2877}$ | $\underline{5.5775}$ | **0.1474** | **0.9195** |
> | Inv.+DPS |1.3857 | $\underline{0.2291}$ | 0.2661 | 5.2690 | 0.3078 | 0.8516 |
> | Inv.+FreeDoM |1.3598 | 0.2117 | 0.2497 | 5.0881 | 0.4308 | 0.7742 |
> | Inv.+TFG |1.4471 | 0.2283 | 0.2654 | 5.2346 | 0.3164 | 0.8460 |
> | Ours | $\underline{1.7719}$ | **0.2403** | **0.2962** | **6.0626** | $\underline{0.2251}$ | $\underline{0.8613}$ |
>
> The result exhibits a trend consistent with our previous experiments on SD1.5 (Table 1) and SD3 (Table 7). While Gradient Ascent achieves a high target reward, it suffers from poor HPSv2 and Aesthetic scores, indicating reward hacking. In contrast, our method achieves the high target reward while outperforming all baselines in validation metrics and structural fidelity. This confirms that our control-theoretic approach scales effectively to larger, more powerful generation models.
>
> - **W3:** Missing baselines. Currently, there emerges a number of image editing methods based on flow-matching-based models such as [1,2,3], authors should compare the image editing performance between the proposed method and these baselines.
>
> Unfortunately, it appears that the specific references for [1, 2, 3] were inadvertently omitted from the review text. We would be glad to include discussions or comparisons if the reviewer could provide the specific citations during the discussion period.
>
> That said, we wish to emphasize a distinction regarding the scope of our work; most recent image editing methods for flow-matching models focus on text-driven editing, which relies on the model's pre-learned conditional distribution to map a source image to a target prompt. In contrast, our framework addresses “reward-guided” editing on unconditional generative models. Our goal is to steer the generation toward an external objective defined by any differentiable reward function (e.g., aesthetic scores or classifier logits), a scenario where the target distribution is not necessarily learned by the generative model itself. Therefore, direct comparisons with some text-based editing methods are often not applicable.
>
> - **W4:** The qualitative results are not enough and some flaws still exists (for example, in second row in Figure 3, the ears of the cat are not clear in the image produced by the proposed method). Authors are expected to provide more comprehensive qualitative results, which are important to evaluate the performance.
>
> We appreciate the reviewer's detailed observation. To address the concern regarding the sufficiency of qualitative evaluation, we have provided extensive additional qualitative results for all four image editing tasks in Figure 9 of the revised manuscript. Also, as evidenced by the expanded qualitative samples and the quantitative user study (Table 5), our method consistently achieves a superior balance between reward maximization and structural preservation compared to baselines.

---

### Official Review · Reviewer_oo4y · 2025-10-30

**Soundness:** 3
**Presentation:** 3
**Contribution:** 2
**Rating:** 4
**Confidence:** 3

**Summary:**

This paper introduces a novel, training-free framework for reward-guided image editing by reformulating the process as a trajectory optimal control problem. Existing methods (like DPS or FreeDoM) calculate the gradient of the final reward $r(x_1)$ with respect to the posterior mean $\hat{x}_{1|t}$. They then push the noisy intermediate image $x_t$ in that direction. The paper's key insight is to treat the entire path from the noisy starting point $x_T$ back to the clean image $x_1$ as a single, adjustable trajectory. The authors use the principles of PMP to develop an iterative adjoint-state update algorithm, achieving good results.

**Strengths:**

1. The formulation is noval and techinically sound, bypassing the issue of using posterior mean for approximation.
2. The method works with pre-trained models (both diffusion and flow-matching) without requiring any model fine-tuning.
3. The writing is easy to follow and the logic flows reasonable.

**Weaknesses:**

1. The proposed method is computationally expensive because it requires a high number of model evaluations.
2. The quality of the output depends on hyperparameters like inversion depth ($T$) and the number of iterations ($N$), which requires careful tuning.

**Questions:**

1. To calculate the gradient through the reward function, I assume it also requires gradient backprop through the decoder and the diffusion model? How much memory cost is it?
2. Is it possible to us RL (e.g., REINFORCE) to adapt the framework to incorporate non-differentiable feedback signals?
3. Since the method is formulated as trajectory optimal control, what guarantees can be provided about finding the global optimum, or are the iterative updates susceptible to finding poor local minima?

---

> ### Author Response · Authors · 2025-11-22
> **Rebuttal from Authors**
>
> We thank the reviewer ``oo4y``'s thoughtful feedback and for considering our work. Below we provide point-by-point responses on your questions and concerns.
>
> - W1: The proposed method is computationally expensive because it requires a high number of model evaluations.
>
> We clarify that while our method involves a higher computational cost than standard sampling, the cost scales linearly with the number of inversion depth $T$ and iterations $N$ similar to inversion-based baselines.
> To validate the practical utility, we analyzed the trade-off between computational efficiency and performance in Figure 4-(b) of the revised version. This analysis demonstrates that even when baselines had used an equivalent computational budget by increasing refinement steps such as $N_{recur}$ and $N_{iter}$, they consistently achieve lower reward maximization and source fidelity than our approach. Furthermore, while increasing guidance steps in baselines often leads to performance degradation and artifacts, our method stably converges toward the optimal trajectory without such deterioration.
>
> - W2: The quality of the output depends on hyperparameters like inversion depth T and the number of iterations N, which requires careful tuning.
>
> We first clarify that our framework effectively reduces hyperparameter complexity compared to state-of-the-art baselines. For instance, TFG requires tuning up to six hyperparameters (including inversion depth $T$ and  $N_{iter}, N_{recur}, \rho_t, \mu_t, \bar{\gamma}$), whereas the performance of our method primarily relies on $T$, number of iterations $N$ and reward weight $w$.
>
> We visualize the behavior of our model according to $T$ and $N$ in Figure 6 of the revised version. Once $N$ is sufficient for convergence, the output images are consistent across different $N$ values. Therefore, as discussed in Appendix A.2, $N$ does not require careful tuning since it governs trajectory convergence rather than guidance strength.  Furthermore, the inversion depth $T$ monotonically controls the trade-off between editability and fidelity, offering user controllability rather than exhibiting unstable sensitivity.
>
> - Q1: To calculate the gradient through the reward function, I assume it also requires gradient backprop through the decoder and the diffusion model? How much memory cost is it?
>
> As the reviewer correctly assumed, calculating the gradient of the reward function with respect to $x_t$ requires backpropagation through the decoder and the diffusion model. We note that this is a common requirement shared by all optimization-based baselines, such as DPS, FreeDoM, and TFG. Consequently, the peak GPU VRAM usage for single-image editing on Stable Diffusion 1.5 was ~14.5 GB for both our method and the baselines.
>
> - Q2: Is it possible to use RL (e.g., REINFORCE) to adapt the framework to incorporate non-differentiable feedback signals?
>
> Our proposed method is a training-free optimization algorithm that relies on gradients to update the control signal, without updating the model parameters. Therefore, standard RL approaches like REINFORCE, which involve policy parameter updates, are outside the scope of our current work.
> However, regarding the handling non-differentiable rewards in a training-free manner, derivative-free methods such as Sequential Monte Carlo (SMC) or Beam Search can be effective alternatives [1]. They apply guidance or selection sequentially at each step of the reverse sampling process. In contrast, our method formulates the problem as a single optimal control problem and updates the entire trajectory iteratively.
>
> - Q3: Since the method is formulated as trajectory optimal control, what guarantees can be provided about finding the global optimum, or are the iterative updates susceptible to finding poor local minima?
>
> Thank you for raising an important theoretical point. As noted in the manuscript (L.240-241), PMP provides necessary conditions for optimality. Given the non-convex nature of complex reward landscapes and the high-dimensional image space, our iterative method does not theoretically guarantee finding the global optimum. Finding the true global optimum in such unconstrained landscapes is generally intractable without explicit training on the target distribution.
> However, our key advantage lies in regularization via the control term. Unlike Gradient Ascent, which purely maximizes rewards and thus risks deviating from the data manifold, our formulation explicitly minimizes the control term ($\frac{1}{2}||u||^2$) subject to the generative dynamics. This constraint ensures that even if the converged solution is a local optimum, the resulting trajectory remains faithful to the pre-trained data manifold, yielding high-fidelity and structurally coherent edits that are superior to unconstrained optimization.
>
> ---
>
> [1] “Inference-Time Alignment in Diffusion Models with Reward-Guided Generation: Tutorial and Review”, 2025.

---

> > ### Comment · Reviewer_oo4y · 2025-11-27
> >
> > Thanks the authors for addressing my concerns. I changed my rating to positive.

---

> > > ### Author Response · Authors · 2025-11-27
> > >
> > > We thank again the reviewer ``oo4y`` for thoughtful feedback and consideration, and for raising the score towards acceptance. We are glad that our rebuttal addressed your previous concerns.

---

### Official Review · Reviewer_WApa · 2025-11-01

**Soundness:** 3
**Presentation:** 3
**Contribution:** 2
**Rating:** 4
**Confidence:** 5

**Summary:**

This paper introduces a training-free, reward-guided image editing framework built atop pretrained diffusion and flow-matching models. Instead of relying on adversarial gradients or retraining reward models, the authors reformulate reward-guided editing as a trajectory optimal control problem, where the reverse diffusion trajectory is treated as a controllable path. Using Pontryagin’s Maximum Principle (PMP) and an iterative adjoint-state optimization, the method updates control signals to steer the entire trajectory toward higher rewards while preserving source fidelity.

**Strengths:**

* The paper presents a novel and theoretically grounded formulation of reward-guided editing via stochastic optimal control, bridging generative modeling and control theory.
* The adjoint-based iterative optimization offers a principled alternative to heuristic reward gradients, mitigating adversarial guidance effects.
* Experiments cover diverse tasks and datasets and compare against good baselines..
* The method is model-agnostic, applicable to both diffusion and flow models (via ODE & SDE conversions).
The paper is well written and easy to follow.

**Weaknesses:**

* The proposed iterative adjoint optimization appears computationally demanding. While FreeDoM and DPS operate in O(n), the proposed trajectory-level optimization is $O(n^2)$, limiting scalability.
* A stronger comparison on efficiency–performance trade-offs is needed (e.g., Pareto plots already hinted at in Fig. 4a). It would help to contextualize the computational gains relative to improvement in reward or fidelity.
* It remains unclear how much the improvement arises from multiple refinements versus the control-theoretic formulation itself. Could repeated low-guidance FreeDoM runs approximate similar results?
* Missing analysis of hyperparameter sensitivity, especially the impact of reward weights and guidance terms, both for baselines and the proposed method.
* Grounded text-editing benchmarks (e.g., PIE-Bench [1]) are absent, which limits the community relevance of the text-guided editing evaluation.
* The paper omits discussion of related optimal-control-inspired methods like FlowChef [4], FireFlow [3], and Taming Rectified Flow [2], which are conceptually close in motivation.

[1] “Direct Inversion: Boosting Diffusion-based Editing with 3 Lines of Code,” ICLR 2024.

[2] “Taming Rectified Flow for Inversion and Editing,” ICML 2025.

[3] “FireFlow: Fast Inversion of Rectified Flow for Image Semantic Editing,” ICML 2025.

[4] “FlowChef: Steering of Rectified Flow Models for Controlled Generations,” ICCV 2025.

**Questions:**

* What is the wall-clock or compute comparison (FLOPs or inference time) between the proposed method and FreeDoM/DPS?
* Could the authors provide a Pareto frontier (compute vs. performance) plot across baselines to strengthen the argument of practical utility?
* How does the proposed method relate to ReNO [5], which performs inference-time noise optimization guided by reward models? * Conceptually, is ReNO equivalent to or distinct from gradient ascent in noise space?
* Could the method be extended to VLM-based editing with text prompts? This seems promising given the reward-agnostic formulation.


[5] “ReNO: Enhancing One-step Text-to-Image Models through Reward-based Noise Optimization,” NeurIPS 2024.

---

> ### Author Response · Authors · 2025-11-22
> **Rebuttal from Authors**
>
> We greatly appreciate Reviewer ``WApa`` for the review and thoughtful feedback. Below, we provide detailed point-by-point responses to address your remaining concerns.
>
> - **W1.** The proposed iterative adjoint optimization appears computationally demanding. While FreeDoM and DPS operate in O(n), the proposed trajectory-level optimization is O(n^2), limiting scalability.
>
> We clarify that the computational complexity of our method scales as $O(N \cdot T)$, not $O(T^2)$, where $T$ is the number of diffusion steps and $N$ is the number of optimization iterations. This is analogous to the complexity of FreeDoM and DPS, where their inner optimization loops (often repeated within each denoising step) also introduce a linear multiplier to the total cost. The detailed implementation that ensures this linear $O(n)$ scaling per iteration is provided in Appendix Algorithm 2, 3.
>
> - **W2:** A stronger comparison on efficiency–performance trade-offs is needed (e.g., Pareto plots already hinted at in Fig. 4a). It would help to contextualize the computational gains relative to improvement in reward or fidelity.
>
> We have addressed this by observing the computational cost vs. performance between our method and baselines. Please refer to our detailed response to Question 2 for the proposed inclusion of a Pareto frontier plot.
>
> - **W3:** It remains unclear how much the improvement arises from multiple refinements versus the control-theoretic formulation itself. Could repeated low-guidance FreeDoM runs approximate similar results?
>
> We would like to highlight that the superior performance of our method arises fundamentally from the optimal control-theoretic formulation, not simply from computational overhead or iterative refinement.
> As discussed in Appendix B.2, since the inversion-based method like FreeDoM cannot model the whole reverse trajectory, their refinement relies on stepwise gradients based on the posterior mean approximation. This is often blurry and inaccurate for complex rewards. In contrast, our approach optimizes the control signals by considering the entire trajectory based on the actual terminal reward.
>
> Furthermore, as we demonstrated in Figure 4-(b) in revised version, even when baselines increase refinement steps ($N_{recur}$ and $N_{iter}$), their performance remains lower than ours. This confirms that the performance gain is due to the trajectory-level optimization, rather than computational overhead.
>
>
> - **W4:** Missing analysis of hyperparameter sensitivity, especially the impact of reward weights and guidance terms, both for baselines and the proposed method.
>
> While we demonstrated our method and baselines’ performance on different guidance strengths in Figure 4-(a), we present additional qualitative results in Figure 7 of the revised version to visualize the impact of varying reward weights or guidance scale. Note that the reward weight $w$ in our framework serves as the effective guidance scale, analogous to the gradient scales ($\rho_t, \mu_t$) in baselines. As illustrated, while baselines exhibit high sensitivity and instability (saturation, artifact, etc.) as the guidance scale increases, our method demonstrates a smooth and progressive emphasis on the target reward as the reward weight $w$ increases, maintaining high image quality.
>
>
> - **W5:** Grounded text-editing benchmarks (e.g., PIE-Bench) are absent, which limits the community relevance of the text-guided editing evaluation
>
> We appreciate the suggestion to incorporate standardized benchmarks. We chose to tackle the relatively simple CelebA-HQ task following previous works [1][2] because both our approach and other baselines address a "reward-guided" editing scenario, where samples are optimized solely based on reward values without utilizing any rich information from the text-conditioned image distribution. Regardless of the image editing method, CLIPscore was not sufficiently robust and instructive as text becomes more complex and multi-attribute for general domain editing tasks.
>
> That said, we have conducted additional evaluations on PIE-Bench using the CLIP score as the target reward following our text-guided image editing task.
>
> | Method | [target]CLIP | [valid.]ImageReward | [valid.]HPSv2 | [source]LPIPS | [source]CLIP-I$_{src}$ |
> | --- | :---: | :---: | :---: | :---: | :---: |
> | None | 0.1664 | -1.9179 | 0.1989 |  0.0000 | 1.0000|
> | GA | **0.3904** |  -1.8374 |  $\underline{0.2018}$ |  **0.1250** |  **0.6880** |
> | Inv.+DPS | 0.3099 |  $\underline{-1.8017}$ |  0.1806 |  0.4335 |  0.6216 |
> | Inv.+FreeDoM | 0.3487 |  -1.8187 |  0.1907 |  0.4976 |  0.6068 |
> | Inv.+TFG | 0.3417 |  -1.8285 |  0.1841 |  0.4276 |  0.6120 |
> | Ours |  $\underline{0.3772}$ |  **-1.7247** |  **0.2089** |   $\underline{0.2328}$ |   $\underline{0.6585}$ |
>
> Consistent with the trends observed in Table 4, our method outperforms baselines in maximizing the validation reward(ImageReward, HPSv2) while maintaining superior perceptual quality and source fidelity.

---

> ### Author Response · Authors · 2025-11-22
> **Rebuttal from Authors (continue)**
>
> - **W6:** The paper omits discussion of related optimal-control-inspired methods like FlowChef [5], FireFlow [4], and Taming Rectified Flow [3], which are conceptually close in motivation.
>
> We thank the reviewer for pointing out these highly relevant works. The key distinction lies in both scope and methodology. While the cited works [3, 4, 5] specifically target text-prompt-based editing for Rectified Flow models, our framework addresses a different task of reward-guided editing using arbitrary differentiable objectives across both diffusion (SDE) and flow-matching (ODE) families. Methodologically, unlike [3, 4] which rely on solver-based inversion and attention modulation, or [5] with step-wise guidance requiring user-provided masks for image editing, our approach formulates a formal trajectory optimal control problem. This allows us to optimize the entire generation path via adjoint-state equations, ensuring global structural coherence without relying on attention modulation or user-defined masks. Please refer to a detailed discussion in Appendix C of the revised version.
>
> - **Q1:** What is the wall-clock or compute comparison (FLOPs or inference time) between the proposed method and FreeDoM/DPS?
>
> We’d like to clarify that we have provided the wall-clock time comparisons in Table 8 of the original submission. We include a detailed FLOPs analysis in Table 8 of the revised manuscript. Under the standard hyperparameter settings listed in Table 6, our method incurs approximately 40% more FLOPs and 80% increased inference time compared to TFG. However, as demonstrated in the FLOPs-performance analysis (detailed in our response to Q2), baseline methods fail to achieve comparable reward maximization and source fidelity even when their computational budget is increased.
>
> - **Q2:** Could the authors provide a Pareto frontier (compute vs. performance) plot across baselines to strengthen the argument of practical utility?
>
> To demonstrate that our superior performance is not a result of increased computation, we have added a Pareto frontier analysis in Figure 4-(b) of the revised version, plotting computational budget (FLOPs) against target reward and source image fidelity. Even when baselines utilize additional optimization steps by increasing $N_{recur}$ or $N_{iter}$ to match our computational budget, they consistently yield lower target rewards and worse source preservation than our method. We also observed that excessive guidance steps in FreeDoM and TFG often lead to performance degradation. On the other hand, since increasing iterations $N_{iter}$ in our framework only ensures convergence toward the optimal trajectory, increasing FLOPs in our method improves source fidelity while maximizing the reward.
>
> - **Q3:** How does the proposed method relate to ReNO, which performs inference-time noise optimization guided by reward models? * Conceptually, is ReNO equivalent to or distinct from gradient ascent in noise space?
>
> We thank the reviewer for suggesting the relevant work. Our method differs from ReNO fundamentally in scope and mechanism: while ReNO optimizes the initial noise $\epsilon_T$ specifically for “T2I generation” and is restricted to one-step models to avoid gradient instability during backpropagation, our framework addresses the “image editing” task for general multi-step models (e.g., standard diffusion and flow matching). Instead of optimizing a single initial noise vector, we formulate a trajectory optimal control problem that iteratively updates control signals to steer the entire reverse path of a given source image.
>
>
> - **Q4:** Could the method be extended to VLM-based editing with text prompts? This seems promising given the reward-agnostic formulation.
>
> As the reviewer suggested, the score given by pre-trained VLMs (e.g., text-image alignment) can be directly used in our method as long as the score is differentiable with respect to the input image. As an example, we employed Qwen2.5-VL 3B model to ask the alignment of the given image and the text prompt, and used the next-word probability of “Yes” as a reward. This reward is utilized for a text-guided image editing task on CelebA-HQ, where the result showed a similar trend to Table 4 in the main manuscript.
>
> | Method | [target]Qwen-VL | [valid.]ImageReward | [valid.]HPSv2 | [source]LPIPS | [source]CLIP-I$_{src}$ |
> | -- | :--:|:--:| :--:|:--:| :--: |
> | None | 0.1476 | -0.2404 | 0.2233 | 0.0000 | 1.0000 |
> | GA | **0.8558** | $\underline{-0.3035}$ | $\underline{0.2150}$ | **0.1690** | **0.8601** |
> | Inv.+DPS | 0.6034 | -0.4568 | 0.2031 | 0.3252 | 0.6783 |
> | Inv.+FreeDoM | 0.5789 | -0.3885 | 0.2071 | 0.4692 | 0.5661 |
> | Inv.+TFG | 0.6695 | -0.6559 | 0.1905 | 0.4147 | 0.5831 |
> | Ours | $\underline{0.8207}$ | **0.0922** | **0.2288** | $\underline{0.2707}$ | $\underline{0.7952}$ |
>
> ---
>
> [1] “Flowgrad: Controlling the output of generative odes with gradients”, CVPR 2023.
>
> [2] “TFG: Unified training-free guidance for diffusion models”, NeurIPS 2024

---

### Meta-Review · Area_Chair_Piop · 2026-01-07

**Summary:**

WApa favors rejection. They feel that the approach is theoretically grounded and novel, the paper is clear, and that the method is general purpose. However, they question the computational complexity of the approach,  request more analysis of performance-efficiency trade-offs, and raise concerns about the discussion of prior work. In the rebuttal, the authors point out that the complexity of the method is O(TN) rather than O(T^2), which they claim is close to existing works, add new Pareto plots, add new hyperparameter experiments, and conducting new comparisons (including to VLMs). oo4y initially recommended rejection. They found the approach to be technically sound and general-purpose. They raised concerns about the computational expense and dependence on certain hyperparameters. However, after the rebuttal, they said that they would change their rating to positive. DZiS recommends acceptance. They found the theory to be novel and intuitive. However, they ask about what constraints the differentiability of the loss function places on the method. They also feel that the evaluation has several shortcomings: missing FLUX evaluations, missing recent flow matching baselines, and limited qualitative results. The authors address these issues by performing additional comparisons with FLUX, finding that it performs competitively. HoKy found the formulation to be principled and the theoretical motivation to be sound, and appreciated how the approach avoided common failure modes and applied to many architectures. Like DZiS, they raise concerns about the differentiability requirement. Authors address this by pointing out that this limitation is shared by other gradient-based guidance approaches. They also point out concerns about efficiency. Similar to the answer to WApa, the authors respond to this by saying that the computation is about 80% higher and a Pareto plot. Finally, they raise concerns about the sensitivity to hyperparameters and trajectory initializations. The authors discuss the effect of changing the hyperparameters/initializations in the rebuttal.

The reviews are mixed. After the initial reviews and author response, two reviewers vote for acceptance and two for rejection. Overall the AC feels that the theoretical benefits of the approach outweigh the concerns raised by WApa and HoKy and recommends that the paper be accepted. The authors should address the main points made by the reviewers, including the questions about the architectures raised by DZiS and the questions about limitations and computational issues raised by WApa and HoKy.

**Reviewer Concerns:**

As described above, reviewer WApa and HoKy raised concerns about computational complexity and the speed of the model. These were partially addressed by the authors. They say that the computational complexity grows as O(TN) rather than O(T^2) and provide Pareto plots. Reviewer DZiS's concerns about architectures were partly addressed by new experiments with FLUX (though it is unclear if this would fully address their questions). The questions about differentiability raised by DZiS and HoKy were partially addressed by the authors, which point out that this is a common assumption in the field.

**Reviewer Scores:**

Reviewer oo4y says that they will raise their score. I see no reason that DZiS would reduce their acceptance score, given the authors' answers. It's possible that the two reviewers that favor rejection would consider raising their scores, although the authors only partially addressed their concerns about computation.

---

### Decision · Program_Chairs · 2026-01-26

Accept (Poster)